# Quality-constrained Entropy Maximization Policy Optimization for LLM diversity

## Abstract

Recent research indicates that while alignment methods significantly improve the quality of large language model(LLM) outputs, they simultaneously reduce the diversity of the models' output. Although some methods have been proposed to enhance LLM output diversity, they often come at the cost of reduced performance. In this work, we first theoretically demonstrate that the alignment task can be decomposed into two distributions: quality and diversity. To enhance the diversity of LLM outputs while ensuring quality, we propose the Quality-constrained Entropy Maximization Policy Optimization (QEMPO). QEMPO aims to maximize the output entropy of the policy while ensuring output quality. By adding different constraints to QEMPO, we obtain different policies. To optimize policies, we propose both online and offline training methods. Experiments validate that QEMPO achieves performance comparable to or even better than RLHF while improving output diversity.

## 1 Introduction

Large language models (LLMs) have demonstrated remarkable capabilities in natural language generation, achieving human-level performance in tasks such as language understanding and commonsense reasoning(Yang et al., 2025; Guo et al., 2025; Dubey et al., 2024). Despite their powerful generative abilities, a critical issue has rapidly gained attention among researchers: output diversity (Li et al., 2025; Murthy et al., 2024; Padmakumar & He, 2023; Slocum et al., 2025; Sun et al., 2025; Xu & Zubiaga, 2025; Kirk et al., 2023). Generative diversity refers to the capacity of an LLM to produce text that captures the nuances and variations inherent in human expression. In many applications—particularly those without a single "correct" answer, such as creative writing—it is essential for LLMs to generate a broad range of valid and varied text outputs. A lack of diversity may result in monotonous and homogeneous content, ultimately limiting the models' applicability across downstream tasks.

After pre-training, LLMs typically use alignment methods to enhance their instruction-following capabilities. However, recent research indicates that while alignment methods—including Supervised Fine-Tuning (SFT), Reinforcement Learning from Human Feedback (RLHF)(Ouyang et al., 2022), and Direct Preference Optimization (DPO) (Rafailov et al., 2023)—significantly improve the quality, usefulness, and safety of model outputs, they are widely observed to reduce the diversity of the content generated by LLMs (Li et al., 2025; Slocum et al., 2025; Xu & Zubiaga, 2025). For instance, RLHF tends to produce outputs that are more detectable, lengthy, and repetitive (Xu & Zubiaga, 2025). It has been observed to decrease the entropy of the output distribution (Padmakumar & He, 2023) and flatten conceptual diversity relative to the base model.

In this work, we revisit the roles of diversity and quality in alignment tasks. Proposition 2 demonstrates that alignment tasks can be decomposed into two components: maximizing diversity and maximizing quality. This means that both the quality and diversity of the output should be considered in alignment. Soling entropy maximization or quality maximization may prevent the alignment objective from reaching its optimal state. Although a higher entropy of the policy implies greater diversity in the outputs, exclusively focusing on entropy maximization could lead to a significant degradation in output quality. As shown in Figure 1, policies (a) and (b) have the same entropy; however, the output quality of policy (a) is clearly higher since all its outputs are satisfactory. Conversely, if we pursue only quality maximization, we may obtain a policy with an output distribution like that of policy (c)

in Figure 1. Although all outputs from policy (c) are satisfactory, they might be overly concentrated in a specific region, thereby reducing diversity.

To enhance the diversity of model outputs while ensuring their quality during the alignment process, we propose Quality-constrained Entropy Maximization Policy Optimization (QEMPO). QEMPO aims to maximize the output entropy of the policy while ensuring output quality. By incorporating different constraints, we derive two optimization objectives: QEMPO and QEMPO-KL. We demonstrate the relative magnitudes of the output entropy of the optimal policies under RLHF, QEMPO, and QEMPO-KL under specific conditions. To optimize QEMPO and QEMPO-KL, we propose both online and offline training methods. Experimental results validate the effectiveness of the proposed approaches. Specifically, the contributions of this work are as follows:

- We prove that diversity and quality are two indispensable components in alignment tasks. Furthermore, we prove that Policy Gradient methods primarily optimize only the quality aspect of alignment.
- We establish that the analytical solution to the quality-constrained KL minimization problem shares the same functional form as the optimal policy derived from RLHF.
- We propose QEMPO and QEMPO-KL to enhance policy diversity while maintaining output quality.
- We prove that under specific conditions, the policy derived from QEMPO achieves higher entropy than that of QEMPO-KL, while QEMPO-KL yields higher entropy than the RLHF policy.
- For both QEMPO and QEMPO-KL, we formulate offline and online optimization objectives. Experimental results validate the effectiveness of the proposed methods.

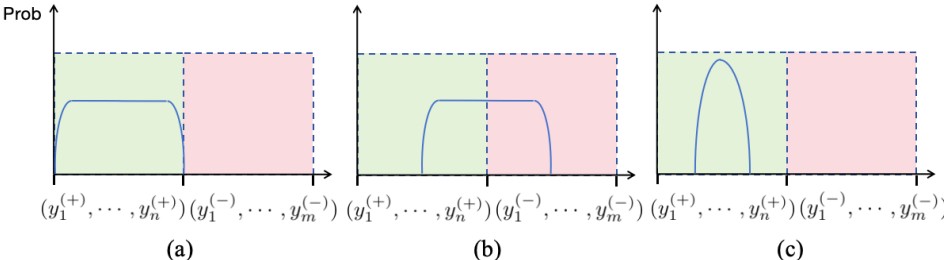

Figure 1: A schematic diagram of the output space distributions of three policies with varying quality and diversity. Here, $(y_1^+, \cdots, y_n^+)$ represents the set of outputs that align with our preferences, while $(y_1^-, \cdots, y_m^-)$ denotes the set of outputs that do not align with our preferences. (a) represents a policy whose outputs exhibit high diversity and very high quality; (b) represents a policy whose outputs exhibit high diversity but lower quality; (c) represents a policy whose outputs have limited diversity but possess high quality.

## 2 PRELIMINARY

### 2.1 RLHF AND DPO

**Notation**. We denote random variables by capital letters and their realizations by lowercase letters. Let $X$ represent the input prompt to the LLM, and $Y$ denote the output. We denote the policy or LLM by $\pi(y|x)$, which can thus be regarded as a distribution. We use $r(x, y)$ to represent the reward model.

**RLHF**. RLHF primarily consists of three steps. First, a model is fine-tuned to serve as the reference policy, denoted as $\pi_{\text{ref}}(x|y)$. Subsequently, human-labeled preference data is collected to train a reward model. Finally, RLHF uses the trained reward model and the reference policy to optimize the following objective:

$$\max_\pi \mathbb{E}_{x \sim D, y \sim \pi}[r(x, y)] - \beta \text{KL}(\pi(y|x) \parallel \pi_{\text{ref}}(y|x)). \quad (1)$$

**DPO**. (Rafailov et al., 2023) first transforms Equation (1) to an equivalent minimization form. Subsequently, by leveraging the non-negativity of the KL divergence, it derives the following closed-form expression for the policy $\pi(x|y)$ at the minimum:

$$\pi(y|x) = \pi_{\text{RLHF}}(y|x) = \frac{1}{Z(x)} \pi_{\text{ref}}(y|x) \exp(\frac{1}{\beta} r(x,y)). \tag{2}$$

Based on the equation above, $r(x,y)$ can be expressed as:

$$r(x,y) = \beta \log \frac{\pi(y|x)}{\pi_{\text{ref}}(y|x)} + \beta \log Z(x). \tag{3}$$

Substituting $r(x,y)$ into the Bradley-Terry model and performing maximum likelihood estimation yields the DPO objective function:

$$\max_{\pi} \mathbb{E}_{y_w \succ y_l \sim D}[\log \sigma(\beta \log \frac{\pi(y_w|x)}{\pi_{\text{ref}}(y_w|x)} - \beta \log \frac{\pi(y_l|x)}{\pi_{\text{ref}}(y_l|x)})] \tag{4}$$

## 2.2 Quality-constrained KL minimization

From a convex optimization perspective, we can view RLHF as an optimization objective that minimizes KL divergence subject to a quality constraint. As can be seen from Proposition 1, when $1/\beta = \lambda$, the analytical solution to the constrained problem in Proposition 1 is identical to that of RLHF. From this, we can observe that although the RLHF optimization objective incorporates quality constraints on the output, it does not inherently encourage diversity in generation. This may lead to RLHF outputs concentrating on only a small subset of satisfactory responses, resulting in a policy as shown in Figure 1-(c).

**Proposition 1.** *For the optimization problem:*

$$\min_{\pi} \text{KL}[\pi(y|x)||\pi_{\text{ref}}(y|x)] \quad s.t. \quad \mathbb{E}_{\pi(y|x)}[r(x,y)] \geq \text{R}$$

*where* $\text{R} = \mathbb{E}\pi_{\text{RLHF}}[r(x,y)]$. *The analytical solution that minimizes this optimization objective is* $\pi(y|x) = \frac{\pi_{\text{ref}}(y|x) \exp(\lambda r(x,y))}{Z(x)}$ *where* $Z(x) = \sum_y \pi_{\text{ref}}(y|x) \exp(\lambda r(x,y))$ *and* $\lambda$ *is the Lagrange multiplier for this optimization objective.*

# 3 Quality-constrained Entropy Maximization Policy Optimization

## 3.1 quality and diversity in Alignment

In this subsection, we revisit the roles of quality and diversity in alignment tasks. We categorize the output into two subsets: $Y^+$ consists of outputs that satisfy the given prompt $X$, while $Y^-$ comprises outputs that do not fulfill $X$. Let $\pi^*(y|x)$ represent our ideal policy. From a practical standpoint, we aim for $\pi^*(y|x)$ to consistently generate outputs that meet the requirements of the given prompt while maximizing output diversity as much as possible. Specifically, $\pi^*(y|x)$ should possess the following two fundamental properties:

**Quality**: $\sum_{y \in Y^+} \pi^*(y|x) = 1 - \epsilon$ where $\epsilon$ is a sufficiently small number.

**Diversity**: $\pi^*(y_i^+ \mid x) = \pi^*(y_j^+ \mid x), \forall y_i^+, y_j^+ \in \text{Y}^+$.

Let $\pi(y|x)$ denote the policy to be aligned. The alignment objective can then be defined as:

$$\min_{\pi} \text{KL}(\pi(y|x)||\pi^*(y|x)). \tag{5}$$

Specifically, we use the entropy of the policy, $H_{\pi}(Y|X)$, to represent diversity, and use $\sum_{y \in Y^+} \pi(y|x)$ to represent the quality of the policy output. From the perspective of information theory, entropy represents the uncertainty of the output, meaning that the probability of the output should not be concentrated in specific areas, which aligns with the goal of diversity. Proposition 2 indicates that to achieve the ideal state of $\pi(y|x)$, both the output diversity and quality of $\pi(y|x)$ should be improved, which also suggests that for alignment tasks, diversity and quality should be considered simultaneously. It is important to note that if we excessively increase the model's output probability

on a small subset of $Y^+$, it is possible to achieve a relatively large value for $\sum_{y \in Y^+} \pi(y|x)$, thereby improving quality. However, in this case, diversity will inevitably decrease, making it impossible to achieve the optimal alignment effect. We further demonstrate that optimization objectives like Policy Gradient (Sutton et al., 1999) inherently only optimize the quality in alignment tasks. A potential drawback of such an optimization objective is that if the policy focuses solely on a single $y^+$ during the learning process, it can still achieve the effect of maximizing the optimization objective, but this is not what we actually desire.

**Proposition 2.** *Let $\pi(y|x)$ denote the policy to be aligned and $\pi^*(y|x)$ the ideal policy. The alignment objective $\mathrm{KL}(\pi(y|x) \parallel \pi^*(y|x))$ can be minimized by jointly maximizing quality and diversity.*

**Corollary 3.1.** *For Policy Gradient, if the reward function is defined as $r(x,y) = \begin{cases} 1 & y \in Y^+ \\ 0 & y \in Y^- \end{cases}$, then Policy Gradient only optimizes the quality component in alignment tasks.*

## 3.2 QUALITY AND KL-CONSTRAINED ENTROPY MAXIMIZATION

Whether from the perspective of the original optimization objective of RLHF or the optimization objective in Proposition 1, RLHF itself does not explicitly encourage diversity in generation. Unlike the optimization objective in Proposition 1, we directly replace the minimizing of KL divergence with maximizing the entropy to explicitly encourage diversity in generation, while treating KL divergence and quality as constraints. Proposition 3 provides the analytical solution form of the corresponding optimization problem. The difference between the analytical solution of RLHF and that of Proposition 3 lies in the addition of an exponential term to $\pi_{\mathrm{ref}}(y|x)$. Since $0 < \frac{\lambda_2}{\lambda_2+1} < 1$, the amplifying effect of the exponential term becomes greater when $\pi_{\mathrm{ref}}(y|x)$ is smaller. For example, $0.9^{0.5} \approx 0.9487$, with an amplification factor of 1.05, while $0.3^{0.5} \approx 0.5477$, with an amplification factor of 1.83. Compared to not adding the exponential term, the effect of $\frac{\lambda_2}{\lambda_2+1}$ is to make the output distribution more uniform. Specifically, Proposition 4 shows that when $\frac{1}{\beta} = \frac{\lambda_1}{\lambda_2}$, $\pi_{\mathrm{QEMPO-KL}}$ has higher entropy than $\pi_{\mathrm{RLHF}}(y|x)$.

**Proposition 3.** *For the optimization problem:* $\max_\pi H_\pi(Y|X)$ *s.t.* $\begin{cases} \mathbb{E}_{\pi(y|x)}[r(x,y)] \geq \mathrm{R} \\ \mathrm{KL}(\pi \parallel \pi_{\mathrm{ref}}) \leq \mathrm{K} \end{cases}$ *where* $\mathrm{R} = \mathbb{E}_{\pi_{\mathrm{RLHF}}}[r(x,y)], \mathrm{K} = \mathrm{KL}(\pi_{\mathrm{RLHF}} \parallel \pi_{\mathrm{ref}})$. *The analytical solution that minimizes this optimization objective is* $\pi_{\mathrm{QEMPO-KL}} = \frac{\pi_{\mathrm{ref}}^{\frac{\lambda_2}{\lambda_2+1}} \exp(\frac{\lambda_1}{\lambda_2+1} r(x,y))}{Z(x)}$ *where $\lambda_1$ and $\lambda_2$ are the Lagrange multipliers for this optimization objective and $Z(x) = \sum_y \pi_{\mathrm{ref}}^{\frac{\lambda_2}{\lambda_2+1}} \exp(\frac{\lambda_1}{\lambda_2+1} r(x,y))$.*

**Proposition 4.** *When $\frac{1}{\beta} = \frac{\lambda_1}{\lambda_2}$, it satisfies:* $H_{\pi_{\mathrm{QEMPO-KL}}}(Y|X) \geq H_{\pi_{\mathrm{RLHF}}}(Y|X)$.

## 3.3 QUALITY-CONSTRAINED ENTROPY MAXIMIZATION

By replacing the optimization objective in Proposition 1 with maximizing policy entropy while treating KL divergence and quality as constraints, we obtain $\pi_{\mathrm{QEMPO-KL}}(y|x)$, which possesses greater entropy than $\pi_{\mathrm{RLHF}}(y|x)$. However, from a constraint perspective, if we remove the KL constraint in Proposition 3, the policy's solution space expands, theoretically implying that the resulting policy should exhibit even greater entropy than $\pi_{\mathrm{QEMPO-KL}}(y|x)$. Moreover, Proposition 2 indicates that quality and diversity are the essential components in alignment tasks. Therefore, we further remove the KL constraint in Proposition 3. Proposition 5 provides the policy $\pi_{\mathrm{QEMPO}}(y|x)$ under the corresponding optimization objective. It can be observed that compared to $\pi_{\mathrm{QEMPO-KL}}(y|x)$, $\pi_{\mathrm{QEMPO}}(y|x)$ removes $\pi_{\mathrm{ref}}(y|x)$. Whether removing $\pi_{\mathrm{ref}}(y|x)$ brings benefits often depends on the quality of $\pi_{\mathrm{ref}}(y|x)$ itself. If the performance of $\pi_{\mathrm{ref}}(y|x)$ is sufficiently high, this implies that the influence of $\pi_{\mathrm{ref}}(y|x)$ on $\pi_{\mathrm{QEMPO-KL}}(y|x)$ can be positive. However, when the performance of $\pi_{\mathrm{ref}}(y|x)$ is inadequate—for instance, when a sample is not the desired one but $\pi_{\mathrm{ref}}(y|x)$ still assigns a high probability—the negative impact on $\pi_{\mathrm{QEMPO-KL}}(y|x)$ can be relatively significant. Nevertheless, from a practical perspective, removing $\pi_{\mathrm{ref}}(y|x)$ reduces memory usage, which is beneficial for the training process. SimPO (Meng et al., 2024) employs the average log probability of a sequence as an implicit reward to eliminate dependency on a reference model. In contrast, QEMPO removes the reference model with the aim of increasing the policy's entropy.

**Proposition 5.** *For the optimization problem:*

$$\max_{\pi} H_{\pi}(Y|X) \quad s.t. \quad \mathbb{E}_{\pi(y|x)}[r(x,y)] \geq \mathrm{R}$$

*where* $\mathrm{R} = \mathbb{E}\pi_{\mathrm{RLHF}}[r(x,y)]$. *The analytical solution that minimizes this optimization objective is* $\pi_{\mathrm{QEMPO}}(y|x) = \frac{\exp(\lambda r(x,y))}{Z(x)}$ *where* $Z(x) = \sum_y \exp(\lambda r(x,y))$ *and* $\lambda$ *is the Lagrange multiplier for this optimization objective.*

**Proposition 6.** *When* $\lambda \leq \frac{\lambda_1}{\lambda_2+1}$ *and* $\frac{\lambda_2}{\lambda_1}\lambda$ *is sufficiently small, it satisfies:* $H_{\pi_{\mathrm{QEMPO}}}(Y|X) \geq H_{\pi_{\mathrm{QEMPO-KL}}}(Y|X)$.

## 3.4 Optimization of QEMPO

In Table 1, we present the forms of the optimal policy solutions under RLHF, QEMPO, and QEMPO-KL. Based on the forms of the optimal policies, we can transform the reward function $r(x,y)$ and derive the corresponding expressions. To optimize QEMPO and QEMPO-KL, we use offline and online modes respectively for optimization.

**Offline mode.** Similarly to DPO, we can substitute the reward expressions corresponding to QEMPO and QEMPO-KL into the BT model to obtain their offline optimization objectives. For QEMPO-KL, the optimization objective is:

$$\max_{\pi} \mathbb{E}_{y^+ \succ y^- \sim D}[\log \sigma(\frac{1}{\lambda_1} \ln \pi(y^+|x) + \frac{\lambda_2}{\lambda_1} \ln \frac{\pi(y^+|x)}{\pi_{\mathrm{ref}}(y^+|x)} - \frac{1}{\lambda_1} \ln \pi(y^-|x) - \frac{\lambda_2}{\lambda_1} \ln \frac{\pi(y^-|x)}{\pi_{\mathrm{ref}}(y^-|x)})]. \tag{6}$$

For QEMPO, the optimization objective is:

$$\max_{\pi} \mathbb{E}_{y^+ \succ y^- \sim D}[\log \sigma(\frac{1}{\lambda} \ln \pi(y^+|x) - \frac{1}{\lambda} \ln \pi(y^-|x))]. \tag{7}$$

**Online mode.** The gradient-weighting method introduced by (Zhang et al., 2025) incorporates reward expressions from optimal policy solutions. By substituting the reward expressions of the QEMPO-KL and QEMPO optimal solutions into this method, we can derive the optimization objectives for QEMPO-KL and QEMPO. For QEMPO-KL, the optimization objective is:

$$2(\frac{\lambda_2+1}{\lambda_1})\mathbb{E}_{x,y}[(r(x,y) - \mathbb{E}[r(x,y)])\log \pi + \frac{\lambda_2}{\lambda_1}\mathrm{Cov}(\log \pi, \log \pi_{\mathrm{ref}}) - 0.5(\frac{\lambda_2+1}{\lambda_1})\mathrm{Var}(\log \pi)] \tag{8}$$

where $r(x,y)$ is the reward function used during training, $\mathrm{Cov}(\log \pi, \log \pi_{\mathrm{ref}})$ denotes the covariance between $\log \pi(y|x)$ and $\log \pi_{\mathrm{ref}}(y|x)$, and $\mathrm{Var}(\log \pi)$ is the variance of $\log \pi(y|x)$. For QEMPO, the optimization objective is:

$$2\frac{1}{\lambda}\mathbb{E}_{x,y}[(r(x,y) - \mathbb{E}[r(x,y)])\log \pi(y|x) - 0.5\frac{1}{\lambda}\mathrm{Var}(\log \pi(y|x))] \tag{9}$$

For both QEMPO and QEMPO-KL in online mode, to ensure quality-constrained policy optimization while maximizing entropy, we optimize entropy only when the policy produces correct responses to questions. Consequently, when policy outputs contain errors, we remove the variance term for such data points in the optimization objective to prevent entropy increase under low-quality outputs.

## 3.5 Discussion

In this subsection, we briefly discuss the distinctions between our method and several existing approaches. Soft Preference Learning (Slocum et al., 2025) decouples the entropy and cross-entropy terms in the KL penalty, allowing for fine-grained control over the diversity of the LLM's output. Unlike SPL, we propose a distinct optimization objective—quality-constrained entropy maximization.

EnTRPO(Roostaie & Ebadzadeh, 2021) combines entropy with the reward value as the optimization objective while incorporating KL divergence as a constraint. ERC-TRPO(Xu et al., 2024) treats the difference between KL divergence and entropy as a constraint and maximizes the reward as the objective. In contrast to both EnTRPO and ERC-TRPO, QEMPO-KL exclusively optimizes for entropy alone, subject to both quality and KL constraints. QEMPO, meanwhile, further simplifies this framework by entirely omitting the KL constraint.

Table 1: The closed-form solution of the optimal policy for RLHF, QEMPO and QEMPO-KL, and the re-expression of the reward function under the optimal policy.

| | Optimal policy | Reward expression |
|---|---|---|
| RLHF | $\pi(y|x) = \frac{\pi_{\text{ref}}(y|x)\exp(\frac{1}{\beta}r(x,y))}{Z(x)}$ | $r_\theta(x,y) = \beta\ln\frac{\pi(y|x)}{\pi_{\text{ref}}(y|x)} + \beta\ln Z(x)$ |
| QEMPO-KL | $\pi(y|x) = \frac{\pi_{\text{ref}}(y|x)^{\frac{\lambda_2}{\lambda_2+1}}\exp(\frac{\lambda_1}{\lambda_2+1}r(x,y))}{Z(x)}$ | $r_\theta(x,y) = \frac{1}{\lambda_1}\ln\pi(y|x) + \frac{\lambda_2}{\lambda_1}\ln\frac{\pi(y|x)}{\pi_{\text{ref}}(y|x)} + \frac{\lambda_2+1}{\lambda_1}\ln Z(x)$ |
| QEMPO | $\pi(y|x) = \frac{\exp(\lambda r(x,y))}{Z(x)}$ | $r_\theta(x,y) = \frac{1}{\lambda}\ln\pi(y|x) + \frac{1}{\lambda}\ln Z(x)$ |

## 4 EXPERIMENTS

### 4.1 EXPERIMENTS IN OFFLINE MODE

#### 4.1.1 SETUP

**Base model.** For smaller models, we used Qwen2.5-1.5B-Instruct (Team, 2024) and Llama-3.2-1B-Instruct (Meta, 2024). For larger models, we used Qwen2.5-7B-Instruct (Team, 2024) and Llama-3.1-8B-Instruct (Dubey et al., 2024).

**Training Data.** We use UltraFeedback Binarized dataset (Cui et al., 2023) for model training. The UltraFeedback Binarized dataset selects higher quality responses as 'chosen' samples, while lower quality responses are labeled as 'rejected' samples.

**Implementation Details in offline mode.** For all experiments, we used a batch size of 128 and trained for 1 epoch. We use Adam (Kingma, 2014) as the optimizer with a warmup ratio of 1. For all experiments involving Llama-3.2-1B-Instruct and Qwen2.5-1.5B-Instruct, the learning rate is set to 5.0e-7. For all experiments involving Llama-3.1-8B-Instruct and Qwen2.5-7B-Instruct, the learning rate is set to 1.0e-7. For RLHF, we follow the setup in (Tunstall et al., 2023) and set $\beta$ to 1e-2. For QEMPO, we set $\frac{1}{\lambda}$ to 4e-3. For QEMPO-KL, we set $\frac{1}{\lambda_1}$ to 4e-3 and set $\frac{\lambda_2}{\lambda_1}$ to 1e-2. We use the test set loss of the trained model on UltraFeedback Binarized as the criterion for selecting the best model.

**Evaluation metrics.** For the quality of LLM outputs, we use gpt-4o-2024-08-06 to evaluate and score it. For the diversity of LLM outputs, we use the diversity evaluation framework proposed by (Guo et al., 2024) to assess it. (Guo et al., 2024) evaluates the diversity of outputs from lexical, syntactic, and semantic dimensions.

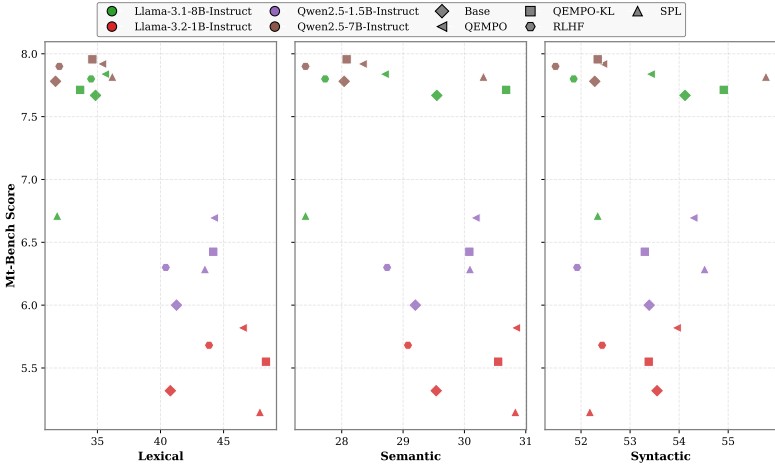

Figure 2: Diversity and quality across different models and methods. Distinct colors represent different base models, while varying shapes denote different approaches.

Table 2: Evaluation results of output diversity and quality on MT-Bench. For diversity, we first sampled each query from the first round of MT-Bench 16 times. Afterward, we randomly selected outputs from the first round as context for the second round and sampled 16 times again. Finally, we calculated the overall output diversity. A higher value indicates greater diversity. For quality We used gpt-4o-2024-08-06 to score the outputs of each model.

|  | Lexical | Semantic | Syntactic | Average | MT-Bench |
|---|---|---|---|---|---|
| Qwen2.5-1.5B-Instruct | 41.27 | 29.20 | 53.39 | 41.29 | 6.00 |
| SPL(Qwen2.5-1.5B-Instruct) | 43.53 | 30.09 | 54.52 | 42.71 | 6.29 |
| RLHF(Qwen2.5-1.5B-Instruct) | 40.42 | 28.74 | 51.92 | 40.36 | 6.30 |
| QEMPO(Qwen2.5-1.5B-Instruct) | **44.23** | **30.18** | **54.29** | **42.90** | **6.54** |
| QEMPO-KL(Qwen2.5-1.5B-Instruct) | 44.18 | 30.08 | 53.30 | 42.52 | 6.43 |
| Llama-3.2-1B-Instruct | 40.78 | 29.54 | 53.55 | 41.29 | 5.32 |
| SPL(Llama-3.2-1B-Instruct) | 47.88 | 30.83 | 52.18 | 43.63 | 5.15 |
| RLHF(Llama-3.2-1B-Instruct) | 43.85 | 29.08 | 52.43 | 41.79 | 5.68 |
| QEMPO(Llama-3.2-1B-Instruct) | 46.51 | **30.84** | **53.95** | 43.77 | **5.82** |
| QEMPO-KL(Llama-3.2-1B-Instruct) | **48.36** | 30.55 | 53.38 | **44.10** | 5.55 |
| Qwen2.5-7B-Instruct | 31.68 | 28.04 | 52.28 | 37.33 | 7.78 |
| SPL(Qwen2.5-1.5B-Instruct) | **36.18** | **30.31** | **55.77** | **40.75** | 7.82 |
| RLHF(Qwen2.5-7B-Instruct) | 31.98 | 27.41 | 51.48 | 36.95 | 7.90 |
| QEMPO(Qwen2.5-7B-Instruct) | 35.37 | 28.34 | 52.45 | 38.72 | 7.92 |
| QEMPO-KL(Qwen2.5-7B-Instruct) | 34.60 | 28.08 | 52.34 | 38.34 | **7.96** |
| Llama-3.1-8B-Instruct | 34.85 | 29.55 | 54.12 | 39.51 | 7.67 |
| SPL(Llama-3.1-8B-Instruct) | 31.81 | 27.41 | 52.34 | 37.19 | 6.71 |
| RLHF(Llama-3.1-8B-Instruct) | 34.50 | 27.73 | 51.85 | 38.03 | 7.80 |
| QEMPO(Llama-3.1-8B-Instruct) | **35.60** | 28.70 | 53.42 | 39.24 | **7.84** |
| QEMPO-KL(Llama-3.1-8B-Instruct) | 33.63 | **30.68** | **54.91** | **39.74** | 7.31 |

## 4.2 PERFORMANCE

We evaluate the effectiveness of the proposed methods on MT-Bench. We use SPL(Slocum et al., 2025) as baseline. The evaluation results, presented in Table 2, indicate that RLHF generally reduces diversity across most scenarios—a finding consistent with the observations reported by (Xu & Zubiaga, 2025). In contrast, both QEMPO and QEMPO-KL improve diversity compared to RLHF in all experiments, demonstrating the efficacy of our proposed approaches. In most cases, SPL also achieves higher output diversity than RLHF, which aligns with the results reported in (Slocum et al., 2025). However, its output diversity is the lowest on Llama-3.1-8B-Instruct, indicating the instability of SPL across different models. Although SPL achieves higher output diversity compared to RLHF, its quality is lower than that of RLHF. In some cases, the results on models like Llama-3.2-1B-Instruct and Llama-3.1-8B-Instruct are even worse than those of the base model. Unlike SPL, QEMPO consistently outperforms RLHF overall. While QEMPO-KL performs comparably to RLHF, it achieves better results on the Qwen model series but slightly underperforms on the Llama series. Figure 2 also reveals that, at similar model sizes, the Llama series exhibits greater diversity than Qwen. This suggests that QEMPO-KL provides more substantial improvements for models with initially lower diversity under comparable scales.

### 4.2.1 ABLATION STUDY

We selected Qwen2.5-1.5B-Instruct to conduct a sensitivity analysis with multiple different hyperparameter settings for both QEMPO and QEMPO-KL. For QEMPO, we set $\frac{1}{\lambda}$ to { 1e-2, 6e-3, 2e-3, 1e-3 } . For QEMPO-KL, we set $(\frac{1}{\lambda_1}, \frac{\lambda_2}{\lambda_1})$ to { (2e-3, 1e-2), (6e-3, 1e-2), (4e-3, 6e-3), (4e-3, 1.2e-2) }. The experimental results are shown in Table 3. It can be observed that while selecting different hyperparameters has some impact on output diversity and quality, the overall variation is not significant. Compared to RLHF's average diversity of 40.36, all results from QEMPO and QEMPO-KL perform better. Moreover, in terms of quality, all results from QEMPO are no lower than RLHF's score of 6.30, while only one result on QEMPO-KL is slightly lower than RLHF.

Table 3: Quality and diversity of Qwen2.5-1.5B-Instruct when selecting different hyperparameters on QEMPO and QEMPO-KL.

|  | Lexical | Semantic | Syntactic | Average | Mt-bench |
|---|---|---|---|---|---|
| QEMPO($\frac{1}{\lambda}$=1e-2) | **45.30** | **30.98** | **54.46** | **43.58** | 6.31 |
| QEMPO($\frac{1}{\lambda}$=6e-3) | 44.78 | 30.48 | 54.44 | 43.23 | 6.34 |
| QEMPO($\frac{1}{\lambda}$=4e-3) | 44.23 | 30.18 | 54.29 | 42.90 | **6.54** |
| QEMPO($\frac{1}{\lambda}$=2e-3) | 42.45 | 29.25 | 53.43 | 41.71 | 6.48 |
| QEMPO($\frac{1}{\lambda}$=1e-3) | 42.51 | 29.74 | 53.11 | 41.79 | 6.30 |
| QEMPO-KL($\frac{1}{\lambda_1}$=4e-3, $\frac{\lambda_2}{\lambda_1}$=1e-2) | 44.18 | 30.08 | 53.30 | 42.52 | **6.43** |
| QEMPO-KL($\frac{1}{\lambda_1}$=2e-3, $\frac{\lambda_2}{\lambda_1}$=1e-2) | 43.20 | 29.45 | 52.93 | 41.86 | 6.35 |
| QEMPO-KL($\frac{1}{\lambda_1}$=4e-3, $\frac{\lambda_2}{\lambda_1}$=6e-3) | 43.95 | 30.15 | 53.26 | 42.45 | 6.36 |
| QEMPO-KL($\frac{1}{\lambda_1}$=4e-3, $\frac{\lambda_2}{\lambda_1}$=1.2e-2) | **44.90** | 29.90 | 52.69 | 42.50 | 6.28 |
| QEMPO-KL($\frac{1}{\lambda_1}$=6e-3, $\frac{\lambda_2}{\lambda_1}$=1e-2) | 44.85 | **30.31** | **53.58** | **42.91** | 6.34 |

### 4.3 EXPERIMENTS IN ONLINE MODE

### 4.4 SETUP

**Base model.** We selected Qwen2.5-7B-Instruct as the base model because its mathematical capabilities are relatively stronger compared to other models.

**Training Data.** We utilized a combined total of 12,000 samples from the training sets of GSM8K(Cobbe et al., 2021) and MATH500(Lightman et al., 2023), each containing a given question and its corresponding answer.

**Implementation Details in online mode.** For all experiments, we set the training batch size to 1024, the number of training epochs to 3, and the learning rate to 1e-7 using a constant learning rate schedule. For each query, we generate 10 responses with the temperature set to 1 during inference. We employ the GRPO method for advantage estimation. For the hyperparameters in RLHF, QEMPO, and QEMPO-KL, our settings are consistent with those in the offline mode. We employ xverify (Chen et al., 2025) to validate whether the reasoning matches the given answer.

**Evaluation metrics.** we employ pass@k to evaluate both the quality and diversity of the model. For the n in pass@k, we set it to 100.

### 4.5 PERFORMANCE

**Quality constraints.** During our experiments, we observed that training QEMPO and QEMPO-KL could lead to an excessive increase in policy entropy, resulting in a sharp decline in reward. We attribute this phenomenon to the fact that entropy increase was encouraged even for low-quality outputs during training. This creates an issue where, when the reward term is zero, the loss can be reduced by increasing entropy. However, increasing entropy under low-quality outputs often leads to further degradation in policy output quality. To prevent over-optimization of entropy for low-quality outputs, we compute the variance term $\mathrm{var}(\log \pi)$ only on data where the outputs are correct. We found that this modification significantly stabilizes the overall training process.

**Performance**. The experimental results are shown in Figure 3. It can be observed that compared to RLHF, QEMPO and QEMPO-KL show little difference in pass@1 across various datasets. However, as k increases, the performance of QEMPO and QEMPO-KL begins to surpass that of RLHF. Moreover, on more challenging datasets, the improvement brought by QEMPO and QEMPO-KL becomes more pronounced as k increases. This indicates that for difficult problems, increasing diversity can also enhance model performance.

## 5 RELATED WORK

Existing research indicates that alignment methods while significantly improving the quality, helpfulness, and safety of LLM outputs, are widely observed to reduce the diversity of the generated content

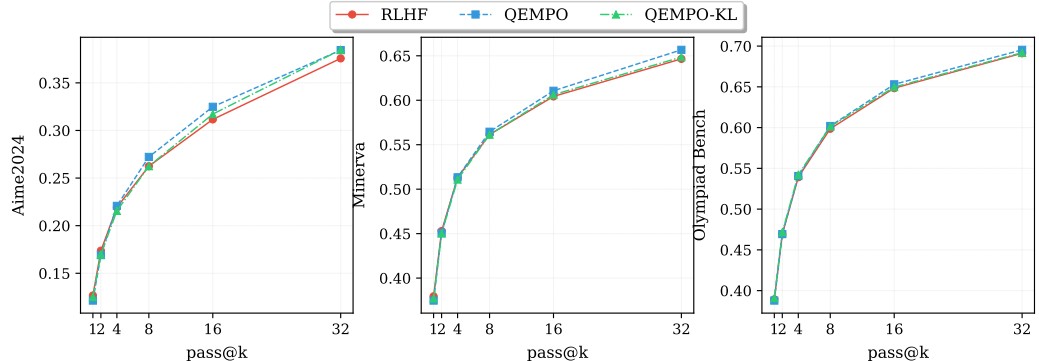

Figure 3: Pass@k results of RLHF, QEMPO, and QEMPO-KL on different datasets.

(Li et al., 2025; Murthy et al., 2024; Padmakumar & He, 2023; Slocum et al., 2025; Sun et al., 2025; Xu & Zubiaga, 2025; Kirk et al., 2023). In response to the decline in diversity attributed to alignment methodologies, researchers have introduced a range of strategies.

**Diversity-oriented data augmentation**. The core idea behind these methodologies centers on enhancing the diversity of large language model (LLM) outputs through the strategic expansion and diversification of training datasets. Research has shown that combining data augmentation with SFT (Wang et al., 2025) and DPO (Lanchantin et al., 2025) can effectively boost the diversity of LLM-generated content.

**Diversity-Oriented Training Methods**. These methods aim to boost the diversity of an LLM's generated output by designing specific reward functions or optimization objectives. Soft Preference Learning (Slocum et al., 2025) decouples the entropy and cross-entropy terms in the KL penalty, allowing for fine-grained control over the diversity of the LLM's output. GEM (Li et al., 2025) uses game theory to design a novel SFT method that preserves the diversity of the model's output. Curiosity-Driven RLHF (Sun et al., 2025) is a framework that combines an intrinsic reward for novel states with the traditional sparse external reward during the RLHF training phase. The trade-off between diversity and quality is examined through the combination of negative log-likelihood training and temperature scaling in (Verine et al., 2025).

**Entropy**. The intrinsic relationship between entropy and the diversity of LLM outputs is direct and significant. Generally, higher entropy in a model's predicted token distribution is associated with greater diversity in the generated text, as it allows the model to explore a wider range of token choices at each step (Sen et al., 2025). Conversely, low-entropy distributions often lead to more deterministic, repetitive, and less diverse outputs (Zhou et al., 2024). However, increasing the entropy of the output distribution can sometimes result in incoherent outputs, thus lowering the quality of the generated text (Carlsson et al., 2024).

## 6 CONCLUSION

In this work, we revisit the roles of quality and diversity in alignment tasks. Theoretically, we demonstrate that quality and diversity are two essential and complementary components of alignment. For RLHF, we prove that the optimal policy form coincides with that of a KL minimization problem subject to quality constraints. To further enhance the entropy of policies under this optimal form, we propose two novel methods: QEMPO and QEMPO-KL. QEMPO is designed to maximize policy entropy under quality constraints, while QEMPO-KL incorporates both quality and KL constraints to maximize entropy. Theoretically, we establish that under certain conditions, the policies derived from QEMPO and QEMPO-KL exhibit higher entropy than those obtained with RLHF. Through both offline and online experiments, we demonstrate that the proposed methods achieve improved diversity while maintaining competitive performance in quality relative to RLHF.

## REPRODUCIBILITY STATEMENT

All theoretical proofs are provided in the appendix. The training data and models designed in this experiment are publicly available. We use trl to implement the offline model experiments for QEMPO and QEMPO-KL. We use verl to implement the online model experiments for QEMPO and QEMPO-KL. To facilitate reproducibility, we provide the following simplified implementation code.

```python
def compute_qempo_kl_loss(beta1, beta2, chosen_logps, rejected_logps,
    ref_chosen_logps, ref_rejected_logps, **kwargs):
    # beta1: \frac{1}{\lambda_{1}} in QEMPO-KL
    # beta2:  \frac{\lambda_{2}}{\lambda_{1}} in QEMPO-KL
    chosen_logratios = beta1 * chosen_logps + beta2 * (chosen_logps -
        ref_chosen_logps)
    rejected_logratios = beta1 * rejected_logps + beta2 * (rejected_logps
        - ref_rejected_logps)
    logits = chosen_logratios - rejected_logratios
    return -F.logsigmoid(logits)
```

Listing 1: A Simple QEMPO-KL Code Implementation in offline mode

```python
def compute_qempo_loss(beta, chosen_logps, rejected_logps, **kwargs):
    # beta: \frac{1}{\lambda} in QEMPO
    chosen_logratios = beta * chosen_logps
    rejected_logratios = beta * rejected_logps
    logits = chosen_logratios - rejected_logratios
    return -F.logsigmoid(logits)
```

Listing 2: A Simple QEMPO Code Implementation in offline mode

```python
def compute_qempo_policy_loss(beta1, beta2, advantages, reward,
    old_log_prob, log_prob, eos_mask, **kwargs):
    # beta1: \frac{1}{\lambda_{1}} in QEMPO-KL
    # beta2:  \frac{\lambda_{2}}{\lambda_{1}} in QEMPO
    score = log_prob - torch.mean(log_prob)
    score_old = old_log_prob - torch.mean(old_log_prob)
    adv = torch.mean(torch.sum(-advantages * log_prob * eos_mask, dim=1))
    cov = beta2 * torch.mean(score * score_old)
    if sum(reward).item() == len(reward_list):
        var = torch.mean(score * score)
    else:
        var = torch.tensor(0.0)
    return 2 * (beta1 + beta2) * (adv + beta2 * cov + (beta1 + beta2) *
        var)
```

Listing 3: A Simple QEMPO-KL Code Implementation within a single group in online mode

```python
def compute_qempo_policy_loss(beta, advantages, reward, log_prob,
    eos_mask, **kwargs):
    # beta: \frac{1}{\lambda} in QEMPO
    score = log_prob - torch.mean(log_prob)
    adv = torch.mean(torch.sum(-advantages * log_prob * eos_mask, dim=1))
    if sum(reward).item() == len(reward_list):
        var = torch.mean(score * score)
    else:
        var = torch.tensor(0.0)
    return 2 * beta * (adv + beta * var)
```

Listing 4: A Simple QEMPO Code Implementation within a single group in online mode

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

APPENDIX

LLM USAGE DECLARATION

In this work, large language models were utilized for two specific purposes: manuscript polishing and partial implementation of experimental code. All conceptual contributions, methodological designs, and result interpretations remain entirely human-authored. The LLM served solely as an implementation aid without substantive intellectual input.

PROOF OF PROPOSITION 1

**Proposition 1.** *For the optimization problem:*

$$\min_\pi \mathrm{KL}[\pi(y|x)||\pi_{\mathrm{ref}}(y|x)] \quad s.t. \quad \mathbb{E}_{\pi(y|x)}[r(x,y)] \geq \mathrm{R}$$

*where* $\mathrm{R} = \mathbb{E}\pi_{\mathrm{RLHF}}[r(x,y)]$. *The analytical solution that minimizes this optimization objective is* $\pi(y|x) = \frac{\pi_{\mathrm{ref}}(y|x)\exp(\lambda r(x,y))}{Z(x)}$ *where* $Z(x) = \sum_y \pi_{\mathrm{ref}}(y|x)\exp(\lambda r(x,y))$ *and* $\lambda$ *is the Lagrange multiplier for this optimization objective.*

*Proof.* We first transform the constraint equivalently, so the original optimization objective can be converted to:

$$\min_\pi \mathrm{KL}[\pi(y|x)||\pi_{\mathrm{ref}}(y|x)] \quad s.t. \quad \mathrm{R} - \mathbb{E}_{\pi(y|x)}[r(x,y)] \leq 0 \tag{10}$$

According to the method of Lagrange multipliers, the following functional can be constructed:

$$L(\pi(y|x)) = \sum_y \pi(y|x)\ln\frac{\pi(y|x)}{\pi_{ref}(y|x)} + \lambda(\mathrm{R} - \sum_y \pi(y|x)r(x,y)) \tag{11}$$

where $\lambda > 0$. Since KL divergence is a convex function for $\pi(y|x)$, its minimum is achieved when the gradient is zero. Setting the derivative of $L(\pi(y|x))$ with respect to $\pi(y|x)$ to zero, we get:

$$\frac{\partial L(\pi(y|x))}{\partial \pi(y|x)} = \ln\pi(y|x) + 1 - \ln\pi_{ref}(y|x) - \lambda r(x,y) = 0. \tag{12}$$

Simplifying the above expression yields:

$$\pi(y|x) = \pi_{\mathrm{ref}}(y|x)\exp(\lambda r(x,y) - 1). \tag{13}$$

Since $\pi(y|x)$ is an LLM, it can be regarded as a distribution. Furthermore, we can obtain:

$$\sum_y \pi(y|x) = \sum_y \pi_{\mathrm{ref}}(y|x)\exp(\lambda r(x,y) - 1) = 1 \tag{14}$$

Finally, we can get:

$$\pi(y|x) = \frac{\pi_{\mathrm{ref}}(y|x)\exp(\lambda r(x,y) - 1)}{1} \tag{15}$$

$$= \frac{\pi_{\mathrm{ref}}(y|x)\exp(\lambda r(x,y) - 1)}{\sum_y \pi_{\mathrm{ref}}(y|x)\exp(\lambda r(x,y) - 1)} \tag{16}$$

$$= \frac{\pi_{\mathrm{ref}}(y|x)\exp(\lambda r(x,y))}{\sum_y \pi_{\mathrm{ref}}(y|x)\exp(\lambda r(x,y))} \tag{17}$$

$$= \frac{\pi_{\mathrm{ref}}(y|x)\exp(\lambda r(x,y))}{Z(x)} \tag{18}$$

where $Z(x) = \sum_y \pi_{\mathrm{ref}}(y|x)\exp(\lambda r(x,y))$. □

PROOF OF COROLLARY PROPOSITION 2

**Proposition 2.** *Let $\pi(y|x)$ denote the policy to be aligned and $\pi^*(y|x)$ the ideal policy. The alignment objective $\mathrm{KL}(\pi(y|x) \parallel \pi^*(y|x))$ can be minimized by jointly maximizing quality and diversity.*

*Proof.* We transform the optimization objective of minimizing KL divergence into a maximization form:

$$\min_\pi \mathrm{KL}(\pi(y|x)||\pi^*(y|x)) = \max_\pi -\mathrm{KL}(\pi(y|x)||\pi^*(y|x)) \tag{19}$$

$$= \max_\pi \underbrace{\sum_y -\pi(y|x)\ln\pi(y|x)}_{H_\pi(Y|X)} + \sum_y \pi(y|x)\ln\pi^*(y|x) \tag{20}$$

For $\sum_y \pi(y|x)\ln\pi^*(y|x)$, we can expand it as:

$$\sum_y \pi(y|x)\ln\pi^*(y|x) = \sum_{y\in Y^+} \pi(y|x)\ln\pi^*(y|x) + \sum_{y\in Y^-} \pi(y|x)\ln\pi^*(y|x). \tag{21}$$

According to the two attributes of $\pi^*(y|x)$—quality and diversity—we can infer that

$$\pi^*(y|x) = \begin{cases} \frac{1-\epsilon}{|Y^+|} & y \in Y^+ \\ \frac{\epsilon}{|Y^-|} & y \in Y^- \end{cases} \tag{22}$$

where $\frac{1-\epsilon}{|Y^+|} \gg \frac{\epsilon}{|Y^-|}$. Furthermore, we can obtain:

$$\sum_y \pi(y|x)\ln\pi^*(y|x) = \sum_{y\in Y^+} \pi(y|x)\ln\frac{1-\epsilon}{|Y^+|} + \sum_{y\in Y^-} \pi(y|x)\ln\frac{\epsilon}{|Y^-|} \tag{23}$$

Let $P_w = \sum_{y\in Y^+} \pi(y|x) \in (0,1)$. Substituting this into the equation above, we obtain:

$$\sum_y \pi(y|x)\ln\pi^*(y|x) = P_w \ln\frac{1-\epsilon}{|Y^+|} + (1-P_w)\ln\frac{\epsilon}{|Y^-|} \tag{24}$$

$$= P_w(\ln\frac{1-\epsilon}{|Y^+|} - \ln\frac{\epsilon}{|Y^-|}) + \ln\frac{\epsilon}{|Y^-|} \tag{25}$$

Since $\ln\frac{1-\epsilon}{|Y^+|} - \ln\frac{\epsilon}{|Y^-|} > 0$, it can be shown that $\sum_y \pi(y|x)\ln\pi^*(y|x)$ increases as $P_w$ increases. Therefore, improving the output quality of $\pi(y|x)$ can increase $\sum_y \pi(y|x)\ln\pi^*(y|x)$, i.e.,

$$\max_\pi \sum_y \pi(y|x)\ln\pi^*(y|x) \propto \max P_w = \max_\pi \sum_{y\in Y^+} \pi(y|x). \tag{26}$$

For diversity, we can use $H_\pi(Y|X)$ to represent it. This is because when the output distribution of $\pi(y|x)$ is more uniform, $H_\pi(Y|X)$ becomes larger.

In summary, the alignment objective $\mathrm{KL}(\pi(y|x)||\pi^*(y|x))$ can be minimized by maximizing both quality and diversity.

$\square$

PROOF OF COROLLARY 3.1

*Proof.* For Policy Gradient, the optimization objective is:

$$\max_\pi \sum_y r(x,y)\pi(y|x). \tag{27}$$

We can further express the above equation as:

$$\max_\pi \sum_{y\in Y} r(x,y)\pi(y|x) = \max_\pi \sum_{y\in Y^+} r(x,y)\pi(y|x) + \max_\pi \sum_{y\in Y^-} r(x,y)\pi(y|x). \tag{28}$$

If the reward function is defined as $r(x,y) = \begin{cases} 1 & y \in Y^+ \\ 0 & y \in Y^- \end{cases}$, then

$$\max_\pi \sum_{y \in Y} r(x,y)\pi(y|x) = \max_\pi \sum_{y \in Y^+} 1 * \pi(y|x) + \max_\pi \sum_{y \in Y^-} 0 * \pi(y|x) \tag{29}$$

$$= \max_\pi \sum_{y \in Y^+} \pi(y|x). \tag{30}$$

Based on the above analysis, we can conclude that Policy Gradient only optimizes the quality component in alignment tasks. $\qquad\square$

## PROOF OF PROPOSITION 3

**Proposition 3.** *For the optimization problem:* $\max_\pi H_\pi(Y|X)$ *s.t.* $\begin{cases} \mathbb{E}_{\pi(y|x)}[r(x,y)] \geq \mathrm{R} \\ \mathrm{KL}(\pi \parallel \pi_{\mathrm{ref}}) \leq \mathrm{K} \end{cases}$ *where* $\mathrm{R} = \mathbb{E}_{\pi_{\mathrm{RLHF}}}[r(x,y)], \mathrm{K} = \mathrm{KL}(\pi_{\mathrm{RLHF}} \parallel \pi_{\mathrm{ref}})$. *The analytical solution that minimizes this optimization objective is* $\pi_{\mathrm{QEMPO-KL}} = \frac{\pi_{\mathrm{ref}}^{\frac{\lambda_2}{\lambda_2+1}} \exp(\frac{\lambda_1}{\lambda_2+1}r(x,y))}{Z(x)}$ *where $\lambda_1$ and $\lambda_2$ are the Lagrange multipliers for this optimization objective and* $Z(x) = \sum_y \pi_{\mathrm{ref}}^{\frac{\lambda_2}{\lambda_2+1}} \exp(\frac{\lambda_1}{\lambda_2+1}r(x,y))$.

*Proof.* We first convert the original optimization problem from maximization to minimization and equivalently transform the constraints, so the original optimization objective can be transformed into:

$$\min_\pi -H_\pi(Y|X) \quad s.t. \begin{cases} \mathrm{R} - \mathbb{E}_{\pi(y|x)}[r(x,y)] \leq 0 \\ \mathrm{KL}(\pi \parallel \pi_{\mathrm{ref}}) - \mathrm{K} \leq 0 \end{cases}. \tag{31}$$

According to the method of Lagrange multipliers, the following functional can be constructed:

$$L(\pi(y|x)) = \sum_y \pi(y|x) \ln \pi(y|x) + \lambda_1(\mathrm{R} - \sum_y \pi(y|x)r(x,y)) + \lambda_2(\sum_y \pi(y|x) \ln \frac{\pi(y|x)}{\pi_{ref}(y|x)} - \mathrm{K}) \tag{32}$$

where $\lambda_1 > 0$ and $\lambda_2 > 0$. Since $-H_\pi(Y|X)$ is a convex function for $\pi(y|x)$, its minimum is achieved when the gradient is zero. Setting the derivative of $L(\pi(y|x))$ with respect to $\pi(y|x)$ to zero, we get:

$$\frac{\partial L(\pi(y|x))}{\partial \pi(y|x)} = \ln \pi(y|x) + 1 - \lambda_1 r(x,y) + \lambda_2 \ln \pi(y|x) + 1 - \lambda_2 \pi_{ref}(y|x) = 0. \tag{33}$$

Simplifying the above equation yields:

$$\pi(y|x) = \pi_{\mathrm{ref}}^{\frac{\lambda_2}{\lambda_2+1}} \exp(\frac{\lambda_1}{\lambda_2+1}r(x,y) - 2). \tag{34}$$

Since $\pi(y|x)$ is an LLM, it therefore satisfies:

$$\sum_y \pi(y|x) = 1. \tag{35}$$

Furthermore, we can obtain:

$$\pi(y|x) = \frac{\pi_{\mathrm{ref}}^{\frac{\lambda_2}{\lambda_2+1}} \exp(\frac{\lambda_1}{\lambda_2+1}r(x,y) - 2)}{1} \tag{36}$$

$$= \frac{\pi_{\mathrm{ref}}^{\frac{\lambda_2}{\lambda_2+1}} \exp(\frac{\lambda_1}{\lambda_2+1}r(x,y) - 2)}{\sum_y \pi_{\mathrm{ref}}^{\frac{\lambda_2}{\lambda_2+1}} \exp(\frac{\lambda_1}{\lambda_2+1}r(x,y) - 2)} \tag{37}$$

$$= \frac{\pi_{\mathrm{ref}}^{\frac{\lambda_2}{\lambda_2+1}} \exp(\frac{\lambda_1}{\lambda_2+1}r(x,y))}{\sum_y \pi_{\mathrm{ref}}^{\frac{\lambda_2}{\lambda_2+1}} \exp(\frac{\lambda_1}{\lambda_2+1}r(x,y))} \tag{38}$$

$$= \frac{\pi_{\mathrm{ref}}^{\frac{\lambda_2}{\lambda_2+1}} \exp(\frac{\lambda_1}{\lambda_2+1}r(x,y))}{Z(x)} \tag{39}$$

where $Z(x) = \sum_y \pi_{\text{ref}}^{\frac{\lambda_2}{\lambda_2+1}} \exp(\frac{\lambda_1}{\lambda_2+1} r(x, y))$. $\qquad\square$

PROOF OF PROPOSITION 4

**Lemma 6.1.** *Let $\mathbf{z} = (z_1, z_2, \ldots, z_n)$ be an arbitrary real-valued vector, and define $p_i(s) = \frac{\exp(sz_i)}{\sum_{j=1}^n \exp(sz_j)} = \frac{\exp(z_i)^s}{\sum_{j=1}^n \exp(z_j)^s}$, and $H(s) = -\sum_{i=1}^n p_i(s) \ln p_i(s)$. Then for any $s_2 > s_1 > 0$, the following holds:*

$$H(s_2) \leq H(s_1),$$

*with equality if and only if all components of $\mathbf{z}$ are equal.*

*Proof.* Let $Z(s) = \sum_{j=1}^n e^{sz_j}$, so that $p_i(s) = \exp(sz_i)/Z(s)$. The derivative of $Z(s)$ with respect to $s$ is:

$$Z'(s) = \sum_{j=1}^n z_j \exp(sz_j). \tag{40}$$

We can compute the derivative of $p_i(s)$ with respect to $s$ as:

$$p_i'(s) = \frac{z_i \exp(sz_i) Z(s) - \exp(sz_i) Z'(s)}{Z(s)^2} \tag{41}$$

$$= \frac{z_i \exp(sz_i)}{Z(s)} - \frac{\exp(sz_i)}{Z(s)} \cdot \frac{Z'(s)}{Z(s)} \tag{42}$$

$$= p_i(s) \left[ z_i - \frac{Z'(s)}{Z(s)}. \right] \tag{43}$$

Note that $\frac{Z'(s)}{Z(s)} = \sum_{j=1}^n z_j p_j(s) = \mathbb{E}_{p(s)}[z]$. Therefore:

$$p_i'(s) = p_i(s) (z_i - \mathbb{E}[z]) \tag{44}$$

where $\mathbb{E}[z] = \sum_j z_j p_j(s)$. The derivative of $H(s)$ with respect to $s$ can be computed as:

$$H'(s) = -\sum_i \left[ p_i'(s) \ln p_i(s) + p_i(s) \cdot \frac{p_i'(s)}{p_i(s)} \right] = -\sum_i p_i'(s) \ln p_i(s) - \sum_i p_i'(s) \tag{45}$$

Note that $\sum_i p_i'(s) = \frac{d}{ds} \sum_i p_i(s) = 0$. Therefore:

$$H'(s) = -\sum_i p_i'(s) \ln p_i(s). \tag{46}$$

Substituting $p_i'(s) = p_i(s)(z_i - \mathbb{E}[z])$ and $\ln p_i(s) = sz_i - \ln Z(s)$ yields:

$$H'(s) = -\sum_i p_i(s)(z_i - \mathbb{E}[z]) [sz_i - \ln Z(s)] \tag{47}$$

$$= -s \sum_i p_i(s)(z_i - \mathbb{E}[z]) z_i + \ln Z(s) \sum_i p_i(s)(z_i - \mathbb{E}[z]). \tag{48}$$

For the expression $\sum_i p_i(s)(z_i - \mathbb{E}[z]) z_i$, we have:

$$\sum_i p_i(s)(z_i - \mathbb{E}[z]) z_i = \sum_i p_i(s)(z_i - \mathbb{E}[z])(z_i - \mathbb{E}[z] + \mathbb{E}[z]) \tag{49}$$

$$= \sum_i p_i(s)(z_i - \mathbb{E}[z])^2 + \mathbb{E}[z] \sum_i p_i(s)(z_i - \mathbb{E}[z]). \tag{50}$$

Since $\sum_i p_i(s)(z_i - \mathbb{E}[z]) = 0$, it follows that:

$$-s \sum_i p_i(s)(z_i - \mathbb{E}[z]) z_i = -s \text{Var}_{p(s)}[z] \tag{51}$$

$$\ln Z(s) \sum_i p_i(s)(z_i - \mathbb{E}[z]) = 0. \tag{52}$$

Therefore, we obtain:

$$H'(s) = -s \cdot \mathrm{Var}_{p(s)}[z]. \tag{53}$$

When not all components of $\mathbf{z}$ are equal, we have $\mathrm{Var}_{p(s)}[z] > 0$. Since $s > 0$, it follows that:

$$H'(s) \leq 0. \tag{54}$$

This implies that $H(s)$ is monotonically decreasing for $s > 0$. Hence, for any $s_2 > s_1 > 0$:

$$H(s_2) < H(s_1). \tag{55}$$

When all components of $\mathbf{z}$ are equal, we have $\mathrm{Var}_{p(s)}[z] = 0$, and therefore:

$$H'(s) = 0. \tag{56}$$

This implies that for any $s_2 > s_1 > 0$:

$$H(s_2) = H(s_1). \tag{57}$$

Combining the above, we complete the proof of the lemma. $\qquad\square$

**Proposition 4.** *When $\frac{1}{\beta} = \frac{\lambda_1}{\lambda_2}$, it satisfies: $H_{\pi_{\mathrm{QEMPO-KL}}}(Y|X) \geq H_{\pi_{\mathrm{RLHF}}}(Y|X)$.*

*Proof.* We first transform $\pi_{\mathrm{QEMPO-KL}}$ as follows:

$$\pi_{\mathrm{QEMPO-KL}}(y|x) = \frac{\pi_{\mathrm{ref}}(y|x)^{\lambda_2/(\lambda_2+1)} \exp(\frac{\lambda_1}{\lambda_2+1} r(x,y))}{Z(x)} \tag{58}$$

$$= \frac{\pi_{\mathrm{ref}}(y|x) \exp(\frac{\lambda_1}{\lambda_2} r(x,y))^{\lambda_2/(\lambda_2+1)}}{Z(x)} \tag{59}$$

$$= \frac{\exp(\ln \pi_{\mathrm{ref}}(y|x) + \frac{\lambda_1}{\lambda_2} r(x,y))^{\lambda_2/(\lambda_2+1)}}{Z(x)}. \tag{60}$$

The expression for $\pi_{\mathrm{RLHF}}(y|x)$ is:

$$\pi_{\mathrm{RLHF}}(y|x) = \frac{\exp(\ln \pi_{ref}(y|x) + \frac{1}{\beta} r(x,y))}{Z(x)}. \tag{61}$$

Since $\lambda_2 > 0$, it follows that $\lambda_2/(\lambda_2 + 1) < 1$. When $\frac{1}{\beta} = \frac{\lambda_1}{\lambda_2}$, applying Lemma 6.1 yields:

$$H_{\pi_{\mathrm{QEMPO-KL}}}(Y|X) \geq H_{\pi_{\mathrm{RLHF}}}(Y|X). \tag{62}$$

$\qquad\square$

## PROOF OF PROPOSITION 5

**Proposition 5.** *For the optimization problem:*

$$\max_\pi H_\pi(Y|X) \quad s.t. \quad \mathbb{E}_{\pi(y|x)}[r(x,y)] \geq \mathrm{R}$$

*where $\mathrm{R} = \mathbb{E}\pi_{\mathrm{RLHF}}[r(x,y)]$. The analytical solution that minimizes this optimization objective is $\pi_{\mathrm{QEMPO}}(y|x) = \frac{\exp(\lambda r(x,y))}{Z(x)}$ where $Z(x) = \sum_y \exp(\lambda r(x,y))$ and $\lambda$ is the Lagrange multiplier for this optimization objective.*

*Proof.* We first convert the original optimization problem from maximization to minimization and equivalently transform the constraints, so the original optimization objective can be transformed into:

$$\min_{\pi} -H_{\pi}(Y|X) \quad s.t. \quad \mathrm{R} - \mathbb{E}_{\pi(y|x)}[r(x,y)] \leq 0 \tag{63}$$

According to the method of Lagrange multipliers, the following functional can be constructed:

$$L(\pi(y|x)) = \sum_{y} \pi(y|x) \ln \pi(y|x) + \lambda(\mathrm{R} - \sum_{y} \pi(y|x)r(x,y)) \tag{64}$$

where $\lambda > 0$. Since $-H_{\pi}(Y|X)$ is a convex function for $\pi(y|x)$, its minimum is achieved when the gradient is zero. Setting the derivative of $L(\pi(y|x))$ with respect to $\pi(y|x)$ to zero, we get:

$$\frac{\partial L(\pi(y|x))}{\partial \pi(y|x)} = \ln \pi(y|x) + 1 - \lambda r(x,y) = 0. \tag{65}$$

Simplifying the above equation yields:

$$\pi(y|x) = \exp(\lambda r(x,y) - 1). \tag{66}$$

Since $\pi(y|x)$ is an LLM, it therefore satisfies:

$$\sum_{y} \pi(y|x) = 1. \tag{67}$$

Furthermore, we can obtain:

$$\pi(y|x) = \frac{\exp(\lambda r(x,y) - 1)}{1} \tag{68}$$

$$= \frac{\exp(\lambda r(x,y) - 1)}{\sum_{y} \exp(\lambda r(x,y) - 1)} \tag{69}$$

$$= \frac{\exp(\lambda r(x,y))}{\sum_{y} \exp(\lambda r(x,y))} \tag{70}$$

$$= \frac{\exp(\lambda r(x,y))}{Z(x)} \tag{71}$$

where $Z(x) = \sum_{y} \exp(\lambda r(x,y))$.

$\square$

## PROOF OF PROPOSITION 6

**Proposition 6.** *When $\lambda \leq \frac{\lambda_1}{\lambda_2+1}$ and $\frac{\lambda_2}{\lambda_1}\lambda$ is sufficiently small, it satisfies: $H_{\pi_{\mathrm{QEMPO}}}(Y|X) \geq H_{\pi_{\mathrm{QEMPO-KL}}}(Y|X)$.*

*Proof.* We first transform $\pi_{\mathrm{QEMPO-KL}}(y|x)$ and $\pi_{\mathrm{QEMPO}}(y|x)$ respectively as follows:

$$\pi_{\mathrm{QEMPO-KL}}(y|x) = \frac{\pi_{\mathrm{ref}}(y|x)^{\lambda_2/(\lambda_2+1)} \exp(\frac{\lambda_1}{\lambda_2+1}r(x,y))}{Z(x)} \tag{72}$$

$$= \frac{\pi_{\mathrm{ref}}(y|x) \exp(\frac{\lambda_1}{\lambda_2}r(x,y))^{\lambda_2/(\lambda_2+1)}}{Z(x)} \tag{73}$$

$$= \frac{\exp(\ln \pi_{\mathrm{ref}}(y|x) + \frac{\lambda_1}{\lambda_2}r(x,y))^{\lambda_2/(\lambda_2+1)}}{Z(x)}. \tag{74}$$

$$\pi_{\mathrm{QEMPO}}(y|x) = \frac{\exp(\lambda r(x,y))}{Z(x)} \tag{75}$$

$$= \frac{\exp(\frac{\lambda_1}{\lambda_2}r(x,y))^{\frac{\lambda_2}{\lambda_1}\lambda}}{Z(x)} \tag{76}$$

When $\frac{\lambda_2}{\lambda_1}\lambda$ is sufficiently small, i.e., $\frac{\lambda_2}{\lambda_1}\lambda \to 0$, we have:

$$\pi_{\text{ref}}(y|x)^{\frac{\lambda_2}{\lambda_1}\lambda} \to 1 \tag{77}$$

At this point, $\pi_{\text{QEMPO}}(y|x)$ is expressed as:

$$\pi_{\text{QEMPO}}(y|x) = \frac{\exp(\ln \pi_{\text{ref}}(y|x) + \frac{\lambda_1}{\lambda_2}r(x,y)))^{\frac{\lambda_2}{\lambda_1}\lambda}}{Z(x)} \tag{78}$$

When $\lambda \le \frac{\lambda_1}{\lambda_2+1}$, it means $\frac{\lambda_2}{\lambda_1}\lambda \le \frac{\lambda_2}{\lambda_2+1}$. Applying Lemma 6.1 yields:

$$H_{\pi_{\text{QEMPO}}}(Y|X) \ge H_{\pi_{\text{QEMPO−KL}}}(Y|X). \tag{79}$$

$\square$

