# OpenReview forum: "Quality-constrained Entropy Maximization Policy Optimization for LLM Diversity"
_ICLR.cc/2026/Conference — Submitted to ICLR 2026_

### Official Review · Reviewer_p75L · 2025-10-15

**Soundness:** 2
**Presentation:** 3
**Contribution:** 2
**Rating:** 2
**Confidence:** 4

**Summary:**

The paper proposes a method that tackles the problem of reduced output diversity in LLM alignment. The proposed approach, QEMPO, directly maximizes the entropy of the output distribution while keeping alignment quality as a constraint.

**Strengths:**

- General writing clarity is good.
- Experimental setups are clearly described, hyperparameters descriptions seem complete, experiments should be reproducible.
- Enough models are used in experiments.

**Weaknesses:**

* **W1) Missing detailed discussion on the limitations of RLHF**. This paper would benefit from a more thorough discussion of why RLHF is not enough to ensure output diversity. I mean it is rather clear after the preliminaries that RLHF never "explicitly" optimizes the entropy of the output distribution. However, a deeper discussion is missing. For example, I do not fully agree with the claim "[RLHF] optimization does not inherently encourage diversity" (line 130) so far. As your optimization perspective in P1 (line 133) suggests, RLHF minimizes KL divergence between the current model and a reference model. Shouldn't this already (implicitly) encourage a diversity close to the one of the reference model? What is the actual reason RLHF has a negative impact on output diversity / entropy? Could the problem also be in the trained reward, e.g. that it does not correctly assign high rewards to alternative responses as well?

* **W2) Unclear diversity definition**. In Section 3.1 you state that the goal of output diversity should be that the probability of all 'correct' sequences must be the same. However, shouldn't the goal of diversity still support responses with varying probabilities? While models might be able to provide alternative responses, why can't some responses be more likely than others? In case your diversity definition is commonly used, could you point to other works that have used this definition before?

* **W3) Weak discussion of experimental results**. In general, the results across models (Table 2) do not consistently suggest that the proposed approach is better than the related work SPL, which also seems to provide better output diversity than RLHF. While this might be okay, a detailed and more careful analysis is missing. For example, what could be reasons for the instability of SPL? How do you ensure the comparison to SPL is fair? Do you tune hyperparameters for the baseline as well? In addition, are your results in Table 2 statistically significant, could you e.g. provide details on the standard deviation (similar for Figure 3)? Overall, the paper would benefit from a more careful discussion of differences to the SPL baseline and of the experimental results in general. This is rather critical for assessing the contribution of this paper.

**Minor weaknesses**

* **Low-quality section 4.2**. While the overall paper is in a good state regarding writing-quality, section 4.2 is repetitive and confusing, sentences are repeated (lines 359-364), making the results difficult to follow.
* **Unclear paragraph in lines 210-212**. I think $\pi_{ref}$ and $\pi_{QEMPO-KL}$ are mixed-up in the discussion, making the argument difficult to understand.
* **Explanation on $\lambda$**. I think it would be helpful to explicitly say that $\lambda$ is the Lagrange multiplier coming from the constraint optimization problem (around Proposition 1 in lines 127-128). Currently this only becomes clear from reading the Appendix. Generally, pointers to proofs in the appendix would also be helpful.
* **Additional references**. Removing the dependency on a reference model (when going from QEMPO-KL to QEMPO in section 3.3) is not new and has been proposed for example by SimPO. I think discussing this parallel would be helpful, including potential negative side-effects (or at least referencing this parallel).
* **Clear preliminaries**. Currently the paper assumes that $\pi$, $x$ and $y$ are clear from context. I think a short introduction in the preliminaries would be helpful to clarify statements such as "Since $\pi$ is an LLM, it can be regarded as a distribution" (L686).
* **Formatting of Table 1**. Table 1 is significantly larger than the page constraints.
* **Typo.** In line 163, do you mean a small subset of $Y_w$ (not $y_w$)?

In general, clearly stating potential limitations and broader impact of the proposed entropy optimization approach would also be helpful.

**Questions:**

- Q1) Could you clarify why entropy maximization under quality constraint is the better perspective? Why is it not sufficient to optimize the RLHF objective under an additional entropy constraint (i.e. the other way around), or simply to maximize both at the same time (RLHF and entropy)? Is QEMPO-KL not just RLHF + entropy maximization after applying the Lagrange multiplier?
- Q2) What do you mean with "policy gradient methods" (e.g. line 006 and in Corollary 3.1 in line 173)? Do you refer to a set of methods? You also write "We further demonstrate that optimization objectives like *Policy Gradient* inherently only optimize the quality in alignment tasks." (line 166). Could you specify what exactly you refer to (and potentially cite it)?
- Q3) How does QEMPO affect general model utility?

---

> ### Author Response · Authors · 2025-11-23
> **Rebuttal by Authors**
>
> Thank you very much for your careful review. We appreciate your feedback and will do our best to answer your questions.
>
> ---
>
> Q: Q1) Could you clarify why entropy maximization under quality constraint is the better perspective? Why is it not sufficient to optimize the RLHF objective under an additional entropy constraint (i.e. the other way around), or simply to maximize both at the same time (RLHF and entropy)? Is QEMPO-KL not just RLHF + entropy maximization after applying the Lagrange multiplier?
>
> A: From a practical application perspective, the output of an LLM must first and foremost meet user expectations to ensure its adoption. Only when output quality is guaranteed can we then focus on enhancing diversity, which is precisely the effect QEMPO aims to achieve. If quality is maximized under entropy or KL constraints, a potential issue is that the policy achieving the highest output quality among all constrained policies may still fail to meet expectations, thereby hindering practical applicability.
>
> SPL [1] decomposes the KL term in RLHF into cross-entropy and entropy, assigning a larger weight coefficient to the entropy term. We can regard SPL as an approach that adds an entropy regularization term to the base RLHF framework. However, both [1] and our experimental results indicate that while SPL enhances diversity, it also leads to a certain degradation in output quality. Therefore, simply maximizing RLHF together with entropy may compromise the quality of the generated outputs.
>
> Since QEMPO(-KL) can be viewed as maximizing the policy entropy under quality and KL constraints, while RLHF can be seen as minimizing the KL divergence under quality constraints—even if entropy maximization is incorporated—the objectives being optimized are fundamentally different. Consequently, when applying Lagrange multipliers, the forms of their Lagrangian functions are not entirely identical.
>
> [1]Diverse preference learning for capabilities and alignment - ICLR2025
>
> ---
>
> Q: Q2) What do you mean with "policy gradient methods" (e.g. line 006 and in Corollary 3.1 in line 173)? Do you refer to a set of methods? You also write "We further demonstrate that optimization objectives like Policy Gradient inherently only optimize the quality in alignment tasks." (line 166). Could you specify what exactly you refer to (and potentially cite it)?
>
> A: Policy gradient methods, a class of reinforcement learning algorithms, directly maximize the expected return by differentiating a parameterized policy, which in large language models corresponds to directly optimizing the probability distribution over generated tokens rather than indirectly shaping it through value functions.  In general, the optimization objective of policy gradient methods can be expressed as: $\max_{\pi}\sum_{y} r(x,y) \pi(y|x)$. We have added the citation for Policy Gradient in the paper at Line 165. Our Proposition 2 indicates that the alignment task can be decomposed into quality and diversity. What Line 166 aims to illustrate is that if our optimization objective is $\max_{\pi}\sum _{y} r(x,y) \pi(y|x)$, then such methods essentially only optimize for quality. Our Corollary 3.1 provides a more detailed theoretical proof for this.
>
> ---
>
> Q: Q3) How does QEMPO affect general model utility?
>
> We evaluated the performance on MMLU of the models trained with QEMPO and QEMPO-KL in Section 4.1. The evaluation results are shown below. It can be observed that QEMPO(-KL) does not negatively affect general model utility.
> ||MMLU|
> | ------------- | --------- |
> |LLama3-8B-Instruct |0.6631|
> |LLama3-8B-Instruct（QEMPO）|0.6642|
> |LLama3-8B-Instruct（QEMPO-KL）|0.6637|
> |Llama-3.2-1B-Instruct|0.4621|
> |Llama-3.2-1B-Instruct（QEMPO）|0.4623|
> |Llama-3.2-1B-Instruct（QEMPO-KL）|0.4628|
> |Qwen2.5-7B-Instruct|0.7412|
> |Qwen2.5-7B-Instruct(QEMPO)|0.7416|
> |Qwen2.5-7B-Instruct(QEMPO-KL)|0.7414|
> |Qwen2.5-1.5B-Instruct|0.6046|
> |Qwen2.5-1.5B-Instruct(QEMPO）|0.6053|
> |Qwen2.5-1.5B-Instruct(QEMPO-KL）|0.6049|

---

> ### Author Response · Authors · 2025-11-23
> **Rebuttal by Authors-part2**
>
> Q: W1) Missing detailed discussion on the limitations of RLHF. This paper would benefit from a more thorough discussion of why RLHF is not enough to ensure output diversity. I mean it is rather clear after the preliminaries that RLHF never "explicitly" optimizes the entropy of the output distribution. However, a deeper discussion is missing. For example, I do not fully agree with the claim "[RLHF] optimization does not inherently encourage diversity" (line 130) so far. As your optimization perspective in P1 (line 133) suggests, RLHF minimizes KL divergence between the current model and a reference model. Shouldn't this already (implicitly) encourage a diversity close to the one of the reference model? What is the actual reason RLHF has a negative impact on output diversity / entropy? Could the problem also be in the trained reward, e.g. that it does not correctly assign high rewards to alternative responses as well?
>
> A: Regarding why RLHF does not explicitly encourage diversity in generation, this can be considered from both experimental and theoretical perspectives. As described in our related work, experimental results from [1-5] all indicate that RLHF reduces output diversity. Theoretically, [2] demonstrates that RLHF diminishes output diversity due to the KL divergence regularization term used in the learning algorithm. For detailed theoretical proofs, please refer to Proposition 3.1 in [2]. To enhance the diversity of RLHF, SPL [2] decouples the entropy and cross-entropy terms in the KL penalty and assigns a larger coefficient to the entropy term to improve output diversity, but this leads to some degradation in output quality. Intuitively, although the KL term includes entropy, it also includes cross-entropy. Adding a KL constraint does not guarantee an increase in the output entropy of the policy, and experimental results generally show that diversity tends to decrease.
>
> [1]Preserving diversity in supervised fine-tuning of large language models - ICLR2025
>
> [2]Diverse preference learning for capabilities and alignment - ICLR2025
>
> [3]Curiosity-driven reinforcement learning from human feedback - arXiv2025
>
>  [4]Understanding the effects of rlhf on the quality and detectability of llm-generated texts - arXiv2025
>
> [5]One fish, two fish, but not the whole sea: Alignment reduces language models’ conceptual diversity - arXiv2024
>
> ---
>
> Q: W2) Unclear diversity definition. In Section 3.1 you state that the goal of output diversity should be that the probability of all 'correct' sequences must be the same. However, shouldn't the goal of diversity still support responses with varying probabilities? While models might be able to provide alternative responses, why can't some responses be more likely than others? In case your diversity definition is commonly used, could you point to other works that have used this definition before?
>
> A: As we described at Line 156, we use the entropy of the policy, $H_{\pi}(Y|X)$, to represent diversity, and use $\sum_{y\in Y^{+}}\pi(y|x)$ to represent the quality of the policy output. From the perspective of information theory, entropy represents the uncertainty of the output, meaning that the probability of the output should not be concentrated in specific areas, which aligns with the goal of diversity. Under theoretical conditions, our definition of diversity can achieve the optimal alignment state. Our Proposition 2 indicates that the alignment task can be decomposed into quality and diversity. If all outputs of a model are "correct" (i.e., quality is maximized) and the LLM assigns equal probability to all "correct" outputs, then the output entropy of the LLM will be maximized. Therefore, to some extent, our definition of diversity aligns with entropy maximization. The use of entropy to enhance output diversity is employed in [1], where the KL regularization is decomposed into entropy and cross-entropy, and the weight coefficient of entropy is increased to improve output diversity. Taking into account Reviewer nhUh's comment, we replace $y_w$ with $y^{+}$ and $y_l$ with $y^{-}$ in the paper to avoid confusion.
>
> [1]Diverse preference learning for capabilities and alignment - ICLR2025

---

> ### Author Response · Authors · 2025-11-23
> **Rebuttal by Authors-part3**
>
> Q: Weak discussion of experimental results. In general, the results across models (Table 2) do not consistently suggest that the proposed approach is better than the related work SPL, which also seems to provide better output diversity than RLHF. While this might be okay, a detailed and more careful analysis is missing. For example, what could be reasons for the instability of SPL? How do you ensure the comparison to SPL is fair? Do you tune hyperparameters for the baseline as well? In addition, are your results in Table 2 statistically significant, could you e.g. provide details on the standard deviation (similar for Figure 3)? Overall, the paper would benefit from a more careful discussion of differences to the SPL baseline and of the experimental results in general. This is rather critical for assessing the contribution of this paper.
>
> A: Consistent with the findings in [1], our experimental results also demonstrate that SPL can generate outputs with better diversity compared to RLHF. However, as indicated by both [1] and our experimental results, the drawback of SPL lies in its trade-off: while it enhances diversity, it compromises output quality.
>
> Unlike SPL, our proposed method aims to maximize entropy under quality constraints. In our experiments, SPL consistently provides better output diversity than RLHF. As described in Section 4.1.1, for the hyperparameters in the SPL algorithm, we used the values provided in [1]. For QEMPO and QEMPO-KL, we employed the same set of hyperparameters across all experiments. All experimental configurations for SPL, such as learning rate and batch size, were kept identical to those of QEMPO.
>
> For Table 2, we have included the standard deviation of the experimental results.
> | | Lexical | Semantic | Syntactic | Avg| MT-Bench|
> | ------------- | --------- | ------------- | --------- | ---------- | --------- |
> |Qwen2.5-1.5B-Instruct | 41.27 ± 0.15 | 29.20 ± 0.11 | 53.39±0.18 | 41.29 | 6.00 ± 0.06|
> |SPL(Qwen2.5-1.5B-Instruct)  | 43.53 ±0.14 | 30.09 ±0.09 | 54.52±0.20 | 42.71  | 6.29±0.06 |
> |RLHF(Qwen2.5-1.5B-Instruct) | 40.42 ± 0.16| 28.74 ±0.10| 51.92±0.18 | 40.36 | 6.30± 0.07|
> |QEMPO(Qwen2.5-1.5B-Instruct) | 44.23± 0.15| 	30.18± 0.10| 	54.29± 0.19| 	42.90 | 	6.54±0.05|
> |QEMPO-KL(Qwen2.5-1.5B-Instruct) | 44.18 ±0.16 | 30.08±0.11 | 53.30± 0.18| 42.52 | 6.43± 0.04|
> |Llama-3.2-1B-Instruct | 40.78 ± 0.16  | 29.54 ±0.13| 53.55±0.21 | 41.29 | 5.32± 0.03|
> |SPL(Llama-3.2-1B-Instruct) | 47.88± 0.15 | 30.83 ±0.15| 52.18± 0.17| 43.63 | 5.15±0.04|
> |RLHF(Llama-3.2-1B-Instruct) | 43.85± 0.16 | 29.08 ±0.10| 52.43± 0.18| 41.79 | 5.68± 0.03|
> |QEMPO(Llama-3.2-1B-Instruct) | 46.51±  0.13| 	30.84 ±0.11| 	53.95±0.19 | 43.77 | 5.82 ±0.03|
> |QEMPO-KL(Llama-3.2-1B-Instruct) | 	48.36±0.16  | 30.55±0.11 | 53.38±0.18 | 	44.10 | 5.55±0.04|
> |Qwen2.5-7B-Instruct | 31.68 ±0.14| 28.04±0.09 | 52.28±0.19 | 37.33 | 7.78 ±0.08|
> |SPL(Qwen2.5-1.5B-Instruct) | 36.18±0.14 | 30.31±0.12 | 55.77±0.16 | 	40.75 |  7.82±0.07|
> |RLHF(Qwen2.5-7B-Instruct) | 31.98 ±0.15| 27.41±0.13 | 51.48± 0.17| 36.95 | 7.90±0.06|
> |QEMPO(Qwen2.5-7B-Instruct) | 35.37± 0.16| 28.34±0.09| 52.45± 0.19| 38.72 | 7.92±0.07|
> |QEMPO-KL(Qwen2.5-7B-Instruct) | 34.60± 0.15| 28.08 ±0.12| 52.34± 0.18| 38.34 |7.96±0.06|
> |Llama-3.1-8B-Instruct | 34.85± 0.17| 29.55 ±0.13| 54.12± 0.15| 39.51 | 7.67 ±0.05|
> |SPL(Llama-3.1-8B-Instruct) | 31.81±0.14 |  27.41±0.11|  52.34±0.15 | 37.19  | 6.71±0.03|
> |RLHF(Llama-3.1-8B-Instruct) | 34.50±0.15 | 27.73 ±0.13| 51.85± 0.17| 38.03 | 7.80 ±0.04|
> |QEMPO(Llama-3.1-8B-Instruct) | 35.60±0.14 | 28.70±0.13 | 53.42± 0.16| 39.24 | 7.84± 0.05|
> |QEMPO-KL(Llama-3.1-8B-Instruct) | 33.63± 0.15|30.68 ±0.10| 54.91±0.15| 39.74| 7.31± 0.04|

---

> ### Author Response · Authors · 2025-11-23
> **Rebuttal by Authors-part4**
>
> Q: Minor weaknesses
>
> 1)Low-quality section 4.2. While the overall paper is in a good state regarding writing-quality, section 4.2 is repetitive and confusing, sentences are repeated (lines 359-364), making the results difficult to follow.
>
> 2)Unclear paragraph in lines 210-212. I think $\pi_{ref}$ and $\pi_{QEMPO-KL}$ are mixed-up in the discussion, making the argument difficult to understand.
>
> 3)Explanation on $\lambda$. I think it would be helpful to explicitly say that $\lambda$ is the Lagrange multiplier coming from the constraint optimization problem (around Proposition 1 in lines 127-128). Currently this only becomes clear from reading the Appendix. Generally, pointers to proofs in the appendix would also be helpful.
>
> 4)Additional references. Removing the dependency on a reference model (when going from QEMPO-KL to QEMPO in section 3.3) is not new and has been proposed for example by SimPO. I think discussing this parallel would be helpful, including potential negative side-effects (or at least referencing this parallel).
>
> 5)Clear preliminaries. Currently the paper assumes that $\pi$, $x$ and $y$ are clear from context. I think a short introduction in the preliminaries would be helpful to clarify statements such as "Since $\pi$ is an LLM, it can be regarded as a distribution" (L686).
>
> 6)Formatting of Table 1. Table 1 is significantly larger than the page constraints.
>
> 7)Typo. In line 163, do you mean a small subset of $Y_w$ (not $y_w$)?
>
> A: We sincerely appreciate your thorough review of our paper. Your suggestions have been immensely helpful in enhancing the readability of this work.
>
> 1)We have removed some redundant and unnecessary sentences in Section 4.2 to make the results clearer.
>
> 2)Thank you very much for your thorough suggestions. We have made the corresponding revisions!
>
> 3)Thank you for your suggestion. We have added an explanation for $\lambda$ in Proposition 1, along with corresponding clarifications in Propositions 3 and 5.
>
> 4)Thank you for your suggestion. We have now included an explanation regarding SimPO in Section 3.3.
>
> 5)Thank you for your suggestion. We have added an explanation regarding the notation in Section 2.1.
>
> 6)Thank you for your suggestion. We have adjusted Table 1 to better fit the page layout.
>
> 7)Yes, we are referring to a small subset of $Y_w$. Thank you for your suggestion; we have revised it accordingly.

---

### Official Review · Reviewer_nhUh · 2025-10-26

**Soundness:** 1
**Presentation:** 1
**Contribution:** 2
**Rating:** 2
**Confidence:** 5

**Summary:**

The paper intends to solve the issue that the diversity of LLMs' outputs after alignment (fine tuning) significantly reduces. It theoretically demonstrates that the alignment task is essentially requiring improving diversity and quality. Then, it proposes a fine-tuning method (with both offline and online versions) such that both quality and diversity can be preserved after fine-tuning.

**Strengths:**

The paper is studying an interesting question.

**Weaknesses:**

1. If I understand correctly, the definitions of $Y_l$ and $Y_w$ are not well-defined.
   - It seems that $Y_l$ and $Y_w$ represent "bad" and "good" responses when deriving theory, where "good" and "bad" are treated as absolute concepts. However, in equation (7) and in section 4.1, $y_w$ and $y_l$ are instead relatively good and bad responses, since the evaluation is based on pairwise comparisons. There seems to be a conceptual gap between these two interpretations, and the paper does not address it clearly.
   - Furthermore, defining $Y_l$ and $Y_w$ is actually nontrivial. If they are meant to include all possible outputs, then the cardinalities $|Y_w|$ and $|Y_l|$ are not guaranteed to be finite. Then a bunch of issues follow. Say, the entire proof of Proposition 2 should be revised. On the other hand, if $Y_w$ and $Y_l$ refer only to subsets of outputs (e.g., within the domain of specific LLMs), then the authors should justify why those particular subsets are considered.
2. The proof of Proposition 2 seems to rely on several assumptions that are not explicitly stated.
   - The **diversity** criterion on page 3 only requires equal probabilities for $Y_w$ but not $Y_l$. However, in the proof of Proposition 2, this condition appears to be applied to both sets.
   - Why does $\frac{1-\varepsilon}{|Y_w|}>\frac{\varepsilon}{|Y_l|}$? This is not obvious since the inequality depends on both values of $\varepsilon$, and $|Y_w|$ and $|Y_l|$. This issue also relates to the problem mentioned in 1.
3. I feel like Proposition 4 and 6 deliver little messages. Since the entropy of a policy depends on hyper parameters, the theoretical results would be more meaningful to include a comparative statement. For example, "when the KL divergence (or expected reward) is fixed, the entropy of XXX is larger than that of XXX."
4. Using Qwen-instruct as base models is somewhat not standard since the Qwen base models already have instructional following capability.
5. The paper's writing quality should be significantly improved. For example, a right quotation mark is utilized where a left quotation mark should be used in the first paragraph. Also, notations are not consistent even on the same page:
   - Proposition 2 uses "KL" while Proposition 3 uses "D_{KL}".
   - In Table 1, some subscripts use "\text{ref}" while some use "\mathrm{ref}".

**Questions:**

See weakness.

---

> ### Author Response · Authors · 2025-11-23
> **Rebuttal by Authors**
>
> Thank you very much for your careful review. We appreciate your feedback and will do our best to answer your questions.
>
> ---
>
> Q: If I understand correctly, the definitions of $Y_l$ and $Y_w$ are not well-defined.
>
> 1)It seems that $Y_l$ and $Y_w$ represent "bad" and "good" responses when deriving theory, where "good" and "bad" are treated as absolute concepts. However, in equation (7) and in section 4.1, $y_w$ and $y_l$ are instead relatively good and bad responses, since the evaluation is based on pairwise comparisons. There seems to be a conceptual gap between these two interpretations, and the paper does not address it clearly.
>
> 2)Furthermore, defining $Y_l$ and $Y_w$ is actually nontrivial. If they are meant to include all possible outputs, then the cardinalities $|Y_w|$ and $|Y_l|$ are not guaranteed to be finite. Then a bunch of issues follow. Say, the entire proof of Proposition 2 should be revised. On the other hand, if $Y_w$ and $Y_l$ refer only to subsets of outputs (e.g., within the domain of specific LLMs), then the authors should justify why those particular subsets are considered.
>
> A: 1)In practical applications, we define the acceptance standard for outputs based on the specific task. If an output meets the standard, i.e., it aligns with your expectations, then it should be treated as an acceptable sample, denoted as $y_w$. For samples that meet the standard, we consider them all to be equally important. Conversely, if an output does not meet the standard, it should be treated as an unacceptable sample, denoted as $y_l$. Thus, our definitions of $y_w$ and $y_l$ are not based on an absolute concept but are relative to the acceptance standard of the specific task. Specifically, we can let $s$ be the task's acceptance threshold score and $s(y)$ be the score assigned to an output sample. Then for $y_w$, we have $s(y_w) \ge s$. For $y_l$, we have $s > s(y_l)$. Therefore, $s(y_w) > s(y_l)$ holds, i.e., $y_w > y_l$. To avoid confusion, we replace $y_w$ with $y^{+}$ and $y_l$ with $y^{-}$ in the paper, while we still retain $y_w$ and $y_l$ in the optimization objective of DPO.
>
> 2)An LLM inherently produces discrete outputs, and in practical applications, we do not allow the LLM to generate infinitely long responses. Theoretically, under a given output length constraint, the model's output is confined to a finite set, namely $V^{L}$, where $V$ is the size of the output vocabulary and $L$ is the output length. Therefore, $|Y_w|$ and $|Y_l|$ are guaranteed to be finite.
>
> ---
>
> Q: The proof of Proposition 2 seems to rely on several assumptions that are not explicitly stated.
>
> The diversity criterion on page 3 only requires equal probabilities for $Y_w$ but not $Y_l$. However, in the proof of Proposition 2, this condition appears to be applied to both sets.
>
> Why does $\frac{1-\varepsilon}{|Y_w|}>\frac{\varepsilon}{|Y_l|}$? This is not obvious since the inequality depends on both values of $\varepsilon$, and $|Y_w|$ and $|Y_l|$. This issue also relates to the problem mentioned in 1.
>
> A: In Section 3.1, we require that the optimal policy $\pi^{\*}$ satisfies both quality and diversity constraints. For **quality**, this implies:
> $\sum_{y_{w}\in Y_{w}}\pi^{\*}(y_{w}|x)  = 1- \epsilon$  and $ \sum_{y_{l}\in Y_{l}}\pi^{\*}(y_{l}|x)  =  \epsilon $   where  $\epsilon$ is a sufficiently small number. Due to the **diversity** requirement, the probability mass is distributed uniformly among all acceptable outputs. Therefore, for any $y_w$, we have: $\pi^{\*}(y_{w} \mid x) = \frac{1 - \epsilon}{|Y_{w}|}$.
>
> For LLM, since $\pi^{\*}(y_{l} \mid x) > 0$ holds for any $y_{l}$, it follows that $\pi^{\*}(y_{l} \mid x) < \epsilon$. Given that all $\pi^{\*}(y_{l} \mid x)$ probabilities are extremely small ($\epsilon$ is a sufficiently small number), we can consider the output probability to be identical for any $y_{l}$. For the optimal policy $\pi^{\*}$, the relation $\pi^{\*}(y_{w} \mid x) > \pi^{\*}(y_{l} \mid x)$ holds, which implies: $\frac{1 - \epsilon}{|Y_{w}|} > \frac{\epsilon}{|Y_{l}|}.$
>
> ---
>
> Q: I feel like Proposition 4 and 6 deliver little messages. Since the entropy of a policy depends on hyper parameters, ...
>
> A: Thank you for your suggestion. However, due to current page length constraints, we have not made this revision for now. We will address it in a subsequent version.
>
> ---
>
> Q: Using Qwen-instruct as base models is somewhat not standard since the Qwen base models already have instructional following capability.
>
> A: We use Qwen-instruct because it has also been employed in some related studies [1, 2].
>
> [1] Benchmarking Linguistic Diversity of Large Language Models
>
> [2] Diversity-oriented Data Augmentation with Large Language Models
>
> ---
>
> Q: The paper's writing quality should be significantly improved. For example, a right quotation mark is utilized where a left quotation mark should be used in the...
>
> A:  Thank you for your suggestions. We have made revisions based on all the recommendations you provided.

---

### Official Review · Reviewer_DWse · 2025-10-29

**Soundness:** 2
**Presentation:** 2
**Contribution:** 2
**Rating:** 4
**Confidence:** 3

**Summary:**

The paper addresses the reduction of output diversity commonly observed in alignment methods such as RLHF and DPO. To mitigate this issue, it introduces Quality-constrained Entropy Maximization Policy Optimization (QEMPO) and its variant QEMPO-KL. QEMPO maximizes the entropy of the policy to enhance diversity while ensuring output quality, whereas QEMPO-KL further constrains the divergence from the reference policy. Analytical expressions for the optimal policies are derived, and both offline (DPO-style) and online (RLHF-style) training objectives are presented. Experiments on Qwen and Llama models show that QEMPO(-KL) improves output diversity metrics while maintaining quality comparable to RLHF.

**Strengths:**

- Clearly motivated by the observed diversity reduction problem in LLM alignment.

- Proposes both offline and online optimization variants (QEMPO, QEMPO-KL) with corresponding closed-form solutions.

- Empirical results on multiple model scales (1B–8B) show consistent diversity improvements across lexical, syntactic, and semantic dimensions.

**Weaknesses:**

The theoretical analysis appears problematic in several parts (although proofs in Appendix were not fully verified).


- Proposition 2 seems to restate the alignment objective as depending on both “quality” and “diversity,” but the result largely follows from how π* is constructed. It is assumed to be uniform over acceptable responses and negligible elsewhere. Under this assumption, the KL term naturally decomposes into an entropy term and a monotone function of the probability mass on Y_w. While mathematically consistent, the statement feels somewhat tautological and informal rather than a general theoretical insight.

- Propositions 3 and 5: The constraints R (minimum quality) and K (maximum KL divergence) are explicitly set to match the values achieved by the RLHF solution. This makes QEMPO(-KL) the maximum-entropy policy among those that already satisfy RLHF’s quality and distance-to-reference levels. While mathematically consistent, it also seems to take RLHF as the implicit reference operating point, so the comparison mainly shows that one can smooth RLHF’s output distribution without violating its constraints. It is therefore not obvious that this constitutes a general superiority over RLHF, rather than an alternative operating point on the same quality/safety frontier.



- (Minor) Reformulating the RLHF KL-regularized objective as a quality-constrained KL minimization problem (Proposition 1) is mathematically correct but essentially a standard Lagrangian dual form. This formulation is well known in the ML literature. Thus, while it provides a convenient foundation for introducing QEMPO, it does not constitute a novel theoretical contribution by itself, even though the authors claim that “we offer a new way to look at the optimization objective of RLHF”.

**Questions:**

- In Proposition 4 and 6, the proof infers higher entropy from pairwise ratio inequalities (Eq. 44). However, such pairwise comparisons do not necessarily imply global entropy increase. Could the authors clarify this point?

- The distinction from prior entropy-regularized RL methods such as EnTRPO and ERC-TRPO seems mainly formal, since both also control reward, KL, and entropy simultaneously. Could the authors clarify what substantive methodological difference or novelty QEMPO(-KL) introduces beyond a different Lagrangian formulation?

**Details Of Ethics Concerns:**

No ethical concerns.

---

> ### Author Response · Authors · 2025-11-23
> **Rebuttal by Authors**
>
> Thank you very much for your careful review. We appreciate your feedback and will do our best to answer your questions.
>
> ---
>
> Q: In Proposition 4 and 6, the proof infers higher entropy from pairwise ratio inequalities (Eq. 44). However, such pairwise comparisons do not necessarily imply global entropy increase. Could the authors clarify this point?
>
> A:  For Proposition 4 and Proposition 6, we provide the following lemmas to facilitate more direct proofs. The lemmas are proved by determining the monotonicity of $H(s)$ through the computation of its derivative. The detailed proofs have been included in the appendix.
>
> **Lemma**: Let $\mathbf{z} = (z_1, z_2, \dots, z_n)$ be an arbitrary real-valued vector, and define  $
> p_i(s) = \frac{\exp({sz_i})}{\sum_{j=1}^n \exp({sz_j})}=\frac{\exp({z_i})^{s}}{\sum_{j=1}^n \exp({z_j})^{s}},$  and  $  H(s) = - \sum_{i=1}^n p_i(s) \ln p_i(s). $   Then for any $s_2 > s_1 > 0$, the following holds: $ H(s_2) \le H(s_1), $ with equality if and only if all components of $\mathbf{z}$ are equal.
>
> The closed-form solution of the optimal policy for RLHF and QEMPO can be expressed as:
>
> $\pi_{\text{RLHF}} (y|x)= \frac{\exp( \ln \pi_{ref}(y|x) +\frac{1}{\beta}r(x,y))}{Z(x)}$
>
> $\pi_{\text{QEMPO-KL}} (y|x)= \frac{ \exp( \ln \pi_{\text{ref}}(y|x)+\frac{\lambda_{1}}{\lambda_{2}} r(x,y))^{\lambda_{2} /(\lambda_{2}+1)}}{Z(x)}$
>
> Since $\lambda_{2}>0$, it follows that $\lambda_{2} /(\lambda_{2}+1)<1$. When $\frac{1}{\beta} =\frac{\lambda_{1}}{\lambda_{2}}$, the lemma provided can be used to prove that the entropy of the QEMPO-KL policy is greater than or equal to that of RLHF. Proposition 6 can also be proven based on this lemma.
>
> ---
>
> Q: The distinction from prior entropy-regularized RL methods such as EnTRPO and ERC-TRPO seems mainly formal, since both also control reward, KL, and entropy simultaneously. Could the authors clarify what substantive methodological difference or novelty QEMPO(-KL) introduces beyond a different Lagrangian formulation?
>
> A: We explain the differences between QEMPO(-KL) and EnTRPO as well as ERC-TRPO from the perspectives of optimization objectives, optimization methods, and the entropy properties of the policy. Regarding the optimization objective, as discussed in Section 3.5, QEMPO(-KL) focuses on maximizing entropy under quality constraints, while EnTRPO and ERC-TRPO treat quality as the optimization goal or one of its components, leading to different Lagrangian formulations. In terms of optimization methods, both EnTRPO and ERC-TRPO employ the TRPO optimization approach, which requires computing the Hessian matrix of the policy. In contrast, our method first derives the closed-form solution of the optimal policy for QEMPO(-KL) and, based on this, designs optimization approaches for both online and offline modes. We prove that under specific conditions, the policy derived from QEMPO and QEMPO-KL yields higher entropy than the RLHF policy. However, no analysis related to the entropy properties of the policies in EnTRPO and ERC-TRPO is provided in [1] and [2].
>
> ---
>
> Q: Proposition 2 seems to restate the alignment objective as depending on both “quality” and “diversity,” but the result largely follows from how π* is constructed. It is assumed to be uniform over acceptable responses and negligible elsewhere. Under this assumption, ...
>
> A: A key point of Proposition 2 is to provide a concrete optimization direction for the policy $\pi$ to be aligned. This is because, in practice, we do not have access to an ideal $π*$. Proposition 2 demonstrates that the alignment objective can be transformed into:
> $ \max_{\pi}\sum_{y\in Y_{w}} \pi(y|x) + \max_{\pi} H_{\pi}(Y|X) $.   Here, our primary focus is on $\pi$. Taking into account Reviewer nhUh's comment, we replace $y_w$ with $y^{+}$ and $y_l$ with $y^{-}$ in the paper to avoid confusion.
>
> ---
>
> Q: Propositions 3 and 5: The constraints R (minimum quality) and K (maximum KL divergence) are explicitly set to match the values achieved by the RLHF solution. This makes QEMPO(-KL) the maximum-entropy policy among those that already satisfy RLHF’s quality and distance-to-reference levels. While mathematically consistent...
>
> A: Our Proposition 2 indicates that the alignment objective can be decomposed into quality and diversity. As implied by your statement, since QEMPO(-KL) becomes the maximum entropy policy among those that satisfy both the RLHF quality level and the reference distance constraint, it inherently means that in terms of the alignment objective, QEMPO(-KL) can always achieve performance that is better than or equal to RLHF.
>
> ---
>
> Q: Minor) Reformulating the RLHF KL-regularized objective as a quality-constrained KL minimization problem (Proposition 1) is mathematically correct but essentially a ...
>
> A: What we aim to illustrate here is that we can view RLHF as an optimization objective that minimizes KL divergence subject to a quality constraint. We have removed this statement to ensure more accurate expression.

---

### Official Review · Reviewer_HwSP · 2025-10-31

**Soundness:** 2
**Presentation:** 2
**Contribution:** 1
**Rating:** 2
**Confidence:** 4

**Summary:**

The paper addresses the growing issue that alignment methods such as RLHF and DPO—while improving the quality, helpfulness, and safety of large language model (LLM) outputs—tend to reduce their diversity. To tackle this, the authors introduce a reinforcement learning framework explicitly designed to preserve or enhance diversity while maintaining quality.

They propose two variants:
- **QEMPO (Quality-constrained Entropy Maximization Policy Optimization):** maximizes the entropy of the model’s output distribution, subject to a *quality constraint*. This encourages more varied responses without sacrificing correctness.
- **QEMPO-KL:** an extension that introduces an additional *KL constraint* relative to a reference policy, controlling divergence from the base model and stabilizing optimization.

Both methods are grounded in a theoretical decomposition of the alignment objective into “quality” and “diversity” components. Analytical derivations show that QEMPO yields the highest entropy, followed by QEMPO-KL and RLHF. The methods are implemented in both offline (DPO-like) and online (policy-gradient-based) training regimes. Empirical results on Qwen and Llama models demonstrate consistent diversity improvements with comparable or slightly better quality scores relative to RLHF.

**Strengths:**

- **Simple and practical modification of RLHF:** The approach extends existing RLHF pipelines with minimal implementation overhead.
- **Code availability:** Implementations for both learning modes are clearly described, supporting reproducibility.

**Weaknesses:**

- **No related works at all:**
  The paper omits nearly all prior work on *quality–diversity trade-offs*, both in evaluation and training, leaving the contribution insufficiently contextualized. This includes several theoretical and empirical studies on balancing diversity and reward, entropy-based evaluation, and diversity-aware optimization methods. These works include dozens of papers, among which I highlight a few important ones and recent contributions below:
  - **Sajjadi, M. S. M. et al.**
  *Assessing Generative Models via Precision and Recall*
  NeurIPS 2018 (NIPS 2018), 2018.
  [arXiv:1806.00035](https://arxiv.org/abs/1806.00035)

   - **Kynkäänniemi, T. et al.**
  *Improved Precision and Recall Metric for Assessing Generative Models*
  NeurIPS 2019 (Advances in Neural Information Processing Systems 32), 2019.
  [arXiv:1904.06991](https://arxiv.org/abs/1904.06991)

   - **Khalifa, M. et al.**
  *A Distributional Approach to Controlled Text Generation*
  ICLR 2021 (camera-ready version), 2020.
  [arXiv:2012.11635](https://arxiv.org/abs/2012.11635)

   - **Le Bronnec, F. et al.**
  *Exploring Precision and Recall to Assess the Quality and Diversity of LLMs*
  ACL 2024 (62nd Annual Meeting of the Association for Computational Linguistics, Long Papers), 2024.
  [arXiv:2402.10693](https://arxiv.org/abs/2402.10693)

   - **Sun, H. et al.**
  *Inverse-RLignment: Large Language Model Alignment from Demonstrations through Inverse Reinforcement Learning*
  arXiv preprint (LLM alignment via inverse RL), 2024.
  [arXiv:2405.15624](https://arxiv.org/abs/2405.15624)

   - **Shypula, A. et al.**
  *Evaluating the Diversity and Quality of LLM Generated Content*
  arXiv preprint (LLM diversity/quality evaluation), 2025.
  [arXiv:2504.12522](https://arxiv.org/abs/2504.12522)

   - **Verine, A. et al.**
  *Improving Diversity in Language Models: When Temperature Fails, Change the Loss*
  ICML 2025 (Forty-Second International Conference on Machine Learning), 2025.
  [arXiv:2508.09654](https://arxiv.org/abs/2508.09654)

   - **Wang, T. et al.**
  *On the Effect of Sampling Diversity in Scaling LLM Inference*
  arXiv preprint (scaling inference / Best-of-N with diverse prompts), 2025.
  [arXiv:2502.11027](https://arxiv.org/abs/2502.11027)


- **Presentation quality:** The manuscript has formatting problems (tables and figures are not properly fitted to the page) and numerous typos, including in the abstract. These issues detract from readability and professionalism.
- **Limited discussion of limitations:** There is little reflection on potential downsides of entropy maximization rather than recall per se, such as incoherent or low-quality generations when constraints are too loose.

**Questions:**

--

---

> ### Author Response · Authors · 2025-11-23
> **Rebuttal by Authors**
>
> We appreciate your feedback and will do our best to answer your questions.
>
> ----
>
> Q: No related works at all: The paper omits nearly all prior work on quality–diversity trade-offs, both in evaluation and training, leaving the contribution insufficiently contextualized. This includes several theoretical and empirical studies on balancing diversity and reward...
>
> A: In fact, we have provided numerous papers related to our research direction in the "Related Work" section. Specifically, [1-7] indicate that alignment methods, while significantly improving the quality, helpfulness, and safety of LLM outputs, are widely observed to reduce the diversity of the generated content. [8, 9] improve the diversity of a large language model's (LLM) output by creating more diverse fine-tuning or preference data for training. [1, 4, 5] boost the diversity of an LLM's generated output by designing specific reward functions or optimization objectives. For the evaluation of LLM diversity, we employ the evaluation metric proposed in [10]. **It is important to note that none of the papers you provided are related to enhancing the diversity of RLHF outputs. Two of the papers you mentioned [11, 12] are entirely unrelated to LLMs, as they focus purely on the field of computer vision.** Therefore, your comment claiming "No related works at all" is completely unfair and irresponsible. We sincerely hope that you can provide more objective comments on our paper. As for some of the works relevant to this paper, we have already included them in the related work section.
>
>
>  [1] Preserving diversity in supervised fine-tuning of large language models. ICLR2025
>
>  [2] One fish, two fish, but not the whole sea: Alignment reduces language models’ conceptual diversity. arXiv2024
>
>  [3] Does writing with language models reduce content diversity? arXiv2023
>
>  [4] Diverse preference learning for capabilities and alignment. ICLR2025
>
>  [5] Curiosity-driven reinforcement learning from human feedback. arXiv2025
>
>  [6] Understanding the effects of rlhf on the quality and detectability of llm-generated texts. arXiv preprint arXiv2025.
>
>  [7] Understanding the effects of rlhf on llm generalisation and diversity. arXiv2023.
>
>  [8] Diversity-oriented data augmentation with large language models. arXiv2025
>
>  [9] Diverse preference optimization. arXiv preprint arXiv2025
>
>  [10] Benchmarking linguistic diversity of large language models. arXiv 2024
>
>  [11] Assessing Generative Models via Precision and Recall NeurIPS 2018
>
>  [12] Improved Precision and Recall Metric for Assessing Generative Models NeurIPS 2019
>
> ---
>
> Q: Presentation quality: The manuscript has formatting problems (tables and figures are not properly fitted to the page) and numerous typos, including in the abstract. These issues detract from readability and professionalism.
>
> A: Regarding the writing of the paper, other reviewers provided specific revision suggestions, and we have made modifications accordingly. **We are very willing to improve the paper's readability if you can offer specific suggestions, as other reviews have done**.
>
> ---
>
> Q: Limited discussion of limitations: There is little reflection on potential downsides of entropy maximization rather than recall per se, such as incoherent or low-quality generations when constraints are too loose.
>
> A: As mentioned in our Introduction, QEMPO(-KL) itself does not solely focus on entropy maximization. Solving entropy maximization or quality maximization in isolation may prevent the alignment objective from reaching its optimal state. Although a higher entropy of the policy implies greater diversity in the outputs, exclusively focusing on entropy maximization could lead to a significant degradation in output quality. In both QEMPO and QEMPO-KL, quality constraints are consistently maintained.

---

> ### Comment · Reviewer_HwSP · 2025-11-26
>
> – I would like to precise the wording *“no related works at all,”* since you indeed have a Related Work section. However, none of the works you cite are **used in comparison** to your model. Meanwhile, several papers that explicitly aim to improve diversity in RLHF — such as [1,2] — should be used as baselines for comparison.
>
> – Regarding your statement that *“two of the papers you mentioned [11,12] are entirely unrelated to LLMs, as they focus purely on computer vision”*: these two papers are indeed originally from computer vision, but they have been **transposed to LLMs** by [13]. More importantly, in these papers you consider unrelated, the **formal definition of quality and diversity** is introduced — the very notions you describe informally in your Figure 1 and in Section 3.1. These works are therefore highly relevant.
>
> – In these papers, *diversity* is measured via **Recall**, i.e. the model’s coverage of the support of the target distribution — which corresponds exactly to **quality-constrained diversity**. This metric is therefore directly related to your work, especially to the distinction between **entropy** and **diversity constrained by quality**, which is central to your proposal.
>
> **References.**
> [1] *Aligning Language Models with Preferences through f-divergence Minimization*, Go et al.
> [2] *Beyond Reverse KL: Generalizing Direct Preference Optimization with Diverse Divergence Constraints*, Wang et al.
> [3] *Inverse-RLignment: Large Language Model Alignment from Demonstrations through Inverse Reinforcement Learning*, Sun et al.
> [11] *Assessing Generative Models via Precision and Recall*, Sajjadi et al.
> [12] *Improved Precision and Recall Metric for Assessing Generative Models*, Kynkäänniemi et al.
> [13] *Exploring Precision and Recall to Assess the Quality and Diversity of LLMs*, Le Bronnec et al.

---

> ### Author Response · Authors · 2025-11-26
> **Rebuttal by Authors-part2**
>
> Q: However, none of the works you cite are used in comparison to your model. Meanwhile, several papers that explicitly aim to improve diversity in RLHF — such as [1,2] — should be used as baselines for comparison.
>
> A: In fact, we clearly state in Section 4.2 that SPL [1] is used as a baseline.
>
> [1] Diverse preference learning for capabilities and alignment. ICLR2025
>
> ---
>
> Q: More importantly, in these papers you consider unrelated, the formal definition of quality and diversity is introduced — the very notions you describe informally in your Figure 1 and in Section 3.1. These works are therefore highly relevant.
>
> – In these papers, diversity is measured via Recall, i.e. the model’s coverage of the support of the target distribution — which corresponds exactly to quality-constrained diversity. This metric is therefore directly related to your work, especially to the distinction between entropy and diversity constrained by quality, which is central to your proposal.
>
> A: In [1], Recall is used to represent Diversity, which is not the same as our definition of Diversity. As we state in Line 156, we use entropy to measure the diversity of the outputs. Note that entropy and Recall are not the same concept. For example, consider a prompt with two correct answers, a1 and a2. Policy A outputs a1 and a2 with confidences 0.1 and 0.9, respectively, while Policy B outputs them with confidences 0.5 and 0.5. In this case, both Policy A and Policy B have a Recall of 1, but their entropies are different.
>
> At the same time, using Recall cannot adequately represent the diversity of LLM outputs. For example, consider a prompt with three correct answers, a1, a2, and a3. Policy A outputs a1, a2, and a3 with confidences 0.001, 0.001, and 0.998, respectively, while Policy B outputs them with confidences 1/3, 1/3, and 1/3. Since Policy A is almost always inclined to produce a3, whereas Policy B can output a1, a2, and a3 with equal probability, Policy B will generate more distinct outputs under the same sampling budget and thus exhibits better output diversity. However, in this case, both Policy A and Policy B achieve a Recall of 1.
>
> [1] Exploring Precision and Recall to Assess the Quality and Diversity of LLMs

---

### Meta-Review · Area_Chair_tFMV · 2025-12-31

**Summary:**

This paper studies the problem that alignment methods such as RLHF and DPO often reduce the diversity of LLM outputs. Motivated by the trade-off of alignment between output quality and diversity, this work proposed Quality-constrained Entropy Maximization Policy Optimization (QEMPO) and its variant QEMPO-KL, which explicitly maximizes the policy entropy under quality constraints. Experiments on Qwen and LLaMA models show consistent improvements in diversity metrics while maintaining comparable output quality.

The main strengths are (1) the research problem of output diversity degradation caused by traditional alignment methods is important, (2) the formulation and solution are straightforward and can be applied to both offline and online methods, (3) the improvement on Qwen and LLaMA models for diversity metrics is consistent. The major concerns are (1) the related works are incomplete, and the novelty of the proposed method is questioned compared to entropy-regularized RL methods, (2) the theoretical contributions are unclear due to the reliance on strong or unclear assumptions and the simple restatement of standard Lagrangian reformulations, (3) the definitions of good/bad response sets, absolute/relative preferences are unclear or inconsistent, (4) the presentation and writing quality has space to be improved. The concerns (see below section) are partially addressed but the presentation is not good enough. The area chair thinks the current version is not ready for publication and encourages the authors to further clarify the novelty, contributions, and theoretical analysis according to reviewers' comments.

**Reviewer Concerns:**

For the concern about insufficient related works, the authors argued that the paper includes numerous related papers, and the papers the reviewer provided are not related to enhancing the diversity of RLHF output. The area chair thinks this concern is well addressed and the papers the reviewer provided are not necessary reasons for rejecting the paper.

For the concern about the novelty compared to entropy-regularized RL methods, the authors explained the differences from multiple perspectives, which makes it clearer to identify the novelty.

For the concerns about definitions, writing quality and presentation, the area chair agrees with reviewers that the current version is not easy to follow and there is space to improve the quality to make the novelty and contributions clearer, considering the weaknesses and questions pointed out in the reviews.

**Reviewer Scores:**

As the clarification on related works and novelty has been addressed, the scores would increase to borderline reject, but considering the presentation and minor concerns, the scores would not be positive.

---

### Decision · Program_Chairs · 2026-01-26

Reject